# Understanding the Logic of Direct Preference Alignment through Logic

Kyle Richardson [1]   Vivek Srikumar [2]   Ashish Sabharwal [1]

## Abstract

Recent direct preference alignment algorithms (DPA), such as `DPO`, have shown great promise in aligning large language models to human preferences. While this has motivated the development of many new variants of the original `DPO` loss, understanding the differences between these recent proposals, as well as developing new DPA loss functions, remains difficult given the lack of a technical and conceptual framework for reasoning about the underlying semantics of these algorithms. In this paper, we attempt to remedy this by formalizing DPA losses in terms of discrete reasoning problems. Specifically, we ask: *Given an existing DPA loss, can we systematically derive a symbolic program that characterizes its semantics?* We propose a novel formalism for characterizing preference losses for single model and reference model based approaches, and identify symbolic forms for a number of commonly used DPA variants. Further, we show how this formal view of preference learning sheds new light on both the size and structure of the DPA loss landscape, making it possible to not only rigorously characterize the relationships between recent loss proposals but also to systematically explore the landscape and derive new loss functions from first principles. We hope our framework and findings will help provide useful guidance to those working on human AI alignment.

## 1. Introduction

Symbolic logic has long served as the de-facto language for expressing complex knowledge throughout computer science (Halpern et al., 2001), including in AI (McCarthy et al., 1960; Nilsson, 1991) and early ML (McCulloch & Pitts, 1943), owing to its clean semantics. Symbolic approaches to

[1]Allen Institute for AI [2]University of Utah. Correspondence to: Kyle Richardson <kyler@allenai.org>.

*Proceedings of the $42^{nd}$ International Conference on Machine Learning*, Vancouver, Canada. PMLR 267, 2025. Copyright 2025 by the author(s).

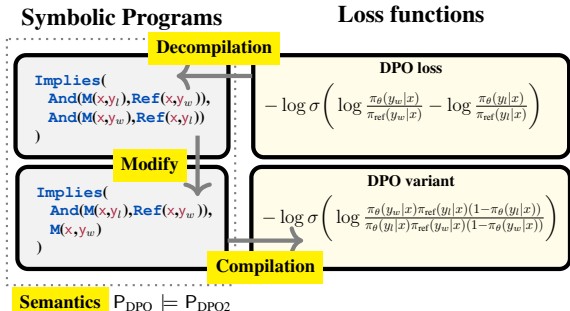

*Figure 1. Can we uncover the hidden logic of* `DPO`*?* Here we show the **decompilation** of the **DPO loss** into a symbolic expression that expresses its high-level model behavior, along with a semantically modified version that we can **compile** into a novel **DPO variant**. We study how to translate between these two spaces to better understand the semantics of existing preference learning algorithms and to derive new ones from first principles.

reasoning that are driven by declarative knowledge, in sharp contrast to purely machine learning-based approaches, have the advantage of allowing us to reason transparently about the behavior and correctness of the resulting systems. In this paper we focus on the broad question: *Can the declarative approach be leveraged to better understand and formally specify algorithms for large language models (LLMs)?*

We specifically investigate **direct preference alignment** (DPA) algorithms, such as direct preference optimization (`DPO`, Rafailov et al., 2023), for pairwise preference learning, which are currently at the forefront of research on LLM alignment and learning from human preferences (Ouyang et al., 2022; Wang et al., 2023). While there has been much recent work on algorithmic variations of `DPO` (Azar et al., 2024; Hong et al., 2024; Meng et al., 2024) that modify or add new terms to the original loss, understanding the differences between these new proposals, as well as coming up with new variants, remains a formidable challenge due to the lack of a conceptual and technical framework for reasoning about their underlying semantics.

Our study attempts to remedy this problem by formalizing the corresponding loss functions in terms of logic, trying to answer the question: *Given an existing loss function, such as* `DPO` *(see Figure 1), can we derive a symbolic expression that captures the core semantics of that loss function (i.e., one that we can then systematically compile back into*

*exactly that same loss)?* By mapping loss functions to discrete reasoning problems — ones that abstract away from lower-level optimization details and reveal high-level model behavior — we can study them using conventional semantic notions from logic (e.g., *entailment*), relate them semantically to other algorithms, or even modify their underlying logical semantics to derive entirely new algorithms. For this formalization, we devise a novel probabilistic logic based on a generalization of the notion of *semantic loss* (SL, Xu et al., 2018) coupled with a provably correct mechanical procedure for translating DPA losses into programs in our logic. As in SL, losses are produced from symbolic programs by counting the weighted propositional models of those programs, reducing the problem to one of probabilistic inference (Chavira & Darwiche, 2008). In contrast to the kinds of symbolic programs commonly used with SL, however, empirically successful DPA losses impose systematic conditional constraints on the types of models that should be counted, which shape the structure of the underlying probability distribution. We express these constraints through a new primitive called a **preference structure** that addresses the technical issues involved with modeling pairwise preference symbolically. Via such constraints, certain semantic relationships between existing losses can be easily observed and new losses can be derived.

Our formal view of preference learning sheds new light on the size and structure of the **DPA loss landscape**. Under modest assumptions motivated by the structure of existing DPA losses, we find that the number of definable preference structures is doubly exponential in the number ($n$) of unique predictions (i.e., forward model calls) made in a loss function, or $4^{2^n}$. This results in an upper bound of 4.3 billion definable DPA losses that are variations of the original DPO loss, leaving much room for exploration. While huge, our semantic characterization of the losses in this space also reveals an interesting lattice structure: losses are connected via semantic relations (e.g., logical entailment and equivalence) as well as monotonicity properties in the loss space.

These formal results also provide practical insights into effectively searching for new DPA losses. For example, one can start with empirically successful loss functions, use the formalization to understand their semantics, then modify their semantics to arrive at novel variants (e.g., more constrained ones), then evaluate. We report on a small-scale case study demonstrating the feasibility of this approach, motivating an exciting avenue for future work.

## 2. Related work

**Language model alignment.** While traditional approaches to language model alignment have employed reinforcement learning (Ziegler et al., 2019; Christiano et al., 2017), we focus on DPA approaches such as DPO (Rafailov et al.,

2023) and SLiC (Zhao et al., 2023) that use closed-form loss functions to tune models directly to offline preferences.

We touch on two recent areas: formal characterizations of DPA losses (Azar et al., 2024; Tang et al., 2024; Hu et al., 2024) and work on devising algorithmically enhanced variants of DPO (Amini et al., 2024; Ethayarajh et al., 2024; Park et al., 2024). In contrast to the former, which focuses on the optimization properties of DPA losses, we attempt to formally characterize the semantic relationships between these variants of DPO in an optimization agnostic way to better understand the DPA loss landscape.

**Neuro-symbolic modeling.** We take inspiration from work on compiling symbolic formulas into novel loss functions (Li et al., 2019; Fischer et al., 2019; Marra et al., 2019; Asai & Hajishirzi, 2020, *inter alia*). We focus particularly on approaches based on probabilistic logic (Manhaeve et al., 2018; Ahmed et al., 2022; 2023a;b; van Krieken et al., 2024b; Calanzone et al., 2025), yet our study differs in focusing on the inverse problem of **decompilation** (see Friedman et al. (2024)), or deriving symbolic expressions from known loss functions (see Appendix A for more related work).

## 3. Direct Preference Alignment

We study offline preference alignment, defined as follows: given data $D_{\mathrm{p}} = \left\{(x^{(i)}, y_w^{(i)}, y_l^{(i)})\right\}_{i=1}^{M}$ consisting of a model input $x$ and two possible generation outputs, a preferred output $y_w$ (the *winner $w$*) and a dispreferred output $y_l$ (the *loser $l$*), the goal is to optimize a policy model (e.g., an LLM) $y \sim \pi_\theta(\cdot \mid x)$ to learn such preferences.

We focus on **direct preference alignment** (DPA) approaches that all take the form of a closed-form loss function $\ell$ that we can use to directly train a model $\pi_\theta$ on $D_{\mathrm{p}}$. Since our study focuses on the formal properties of DPA losses, it is important to understand their general structure, which will take the following form from Tang et al. (2024):

$$\ell_{\mathrm{DPA}}(\theta, D) := \mathop{\mathbb{E}}_{(x, y_w, y_l) \sim D_{\mathrm{p}}} \left[ f\big(\rho_\theta(x, y_w, y_l), \beta\big) \right] \quad (1)$$

consisting of a convex loss function $f : \mathbb{R} \times \mathbb{R}+ \to \mathbb{R}$, a differentiable model quantity $\rho_\theta(x, y_w, y_l)$, which we abbreviate to $\rho_\theta$, and a parameter $\beta$ for scaling terms in $\rho_\theta$.

*Table 1.* Examples of some DPA loss functions (Eq 1) with different choices of convex function $f$ and model quantity $\rho_\theta$.

| | $f(\rho_\theta, \beta) =$ | $\rho_\theta$ |
|---|---|---|
| DPO | $-\log \sigma(\beta \rho_\theta)$ | $\log \frac{\pi_\theta(y_w\mid x)}{\pi_{\mathrm{ref}}(y_w\mid x)} - \log \frac{\pi_\theta(y_l\mid x)}{\pi_{\mathrm{ref}}(y_l\mid x)}$ |
| IPO | $\left(\rho_\theta - \frac{1}{2\beta}\right)^2$ | |
| SLiC | $\max(0, \beta - \rho_\theta)$ | $\log \frac{\pi_\theta(y_w\mid x)}{\pi_\theta(y_l\mid x)}$ |
| RRHF | $\max(0, -\rho_\theta)$ | $\log \frac{\pi_\theta(y_w\mid x)^{\frac{1}{\mid y_w\mid}}}{\pi_\theta(y_l\mid x)^{\frac{1}{\mid y_l\mid}}}$ |

*Table 2. How are variants of* DPO *structured?* We define popular variants in terms of their **core loss equation** $\rho_\theta$ and the helper function $s_{m_1,m_2}(y_1, y_2)$ (last column) that rewrites each $\rho_\theta$ in a way that brings out general shared structural patterns and added terms compared with the log win/loss ratio $s_\theta(y_w, y_l)$. All original losses are implemented via the logistic log loss: $\ell_x = -\log\sigma(\beta\rho_\theta)$.

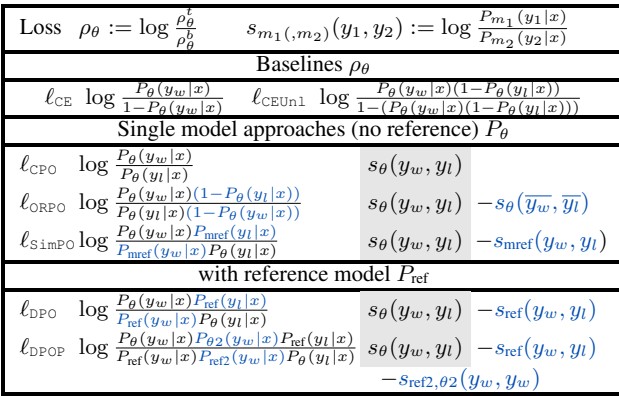

*Figure 2. How do we map losses to discrete reasoning problems?* Our approach and key results (by paper section §). First an **input loss** (upper left) is stripped down to its **core loss equation** (lower left), then **semantically translated** (lower right) and mapped into a semantic structure (upper right) that is provably compilable (under a novel logic, §5) back into the original loss (Thm 1).

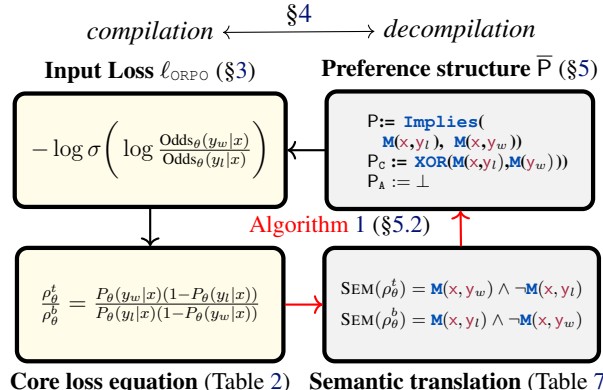

Table 1 lists four specific DPA losses: DPO (Rafailov et al., 2023), IPO (Azar et al., 2024), SliC (Zhao et al., 2022; 2023), and RRHF (Yuan et al., 2023) that differ in $f$ or $\rho_\theta$. Here the logistic log loss (with $\sigma(x) = \frac{1}{1+\exp(-x)}$), square loss, hinge loss, and perceptron loss are used for $f$. SliC and RRHF are examples of **single model** approaches that define $\rho_\theta$ in terms of the **log ratio of the winner and loser** given prediction probabilities of the model being trained $\pi_\theta$. The prediction probabilities are sometimes computed using **length normalization** (i.e., taking a geometric mean of token probabilities) as shown for RRHF. Single model losses are usually regularized using an added cross-entropy term, which we exclude from our formal analysis.[1] For DPO and IPO, in contrast, the model quantity $\rho_\theta$ is the **log ratio difference** (of the winner and the loser) between the predictions of the model being trained and a frozen LLM called a reference model, $\pi_{\text{ref}}$. These two approaches constitute a **two model approach**, where the role of the reference model is to avoid overfitting (controlled by $\beta$).

Conceptually, preference losses involve making predictions about winners and losers across models and reasoning about the relationships between predictions. Our main question is: *If we view this process as a discrete reasoning problem, what is the nature of the reasoning that underlies these different losses and each model quantity $\rho_\theta$?* As shown in Figure 2, our analysis operates at the level of $\rho_\theta$ and starts by rewriting each loss into a core loss equation that strips away optimization/implementation details (e.g., de-

tails about $f$, $\beta$, length normalization). Next we discuss these core loss equations and the general structure of the DPA losses (Table 2) that we aim to derive formally.

**Core loss equations.** Table 2 shows the different variants of DPO we investigate and two common baseline losses from Rafailov et al. (2023) – the cross-entropy loss $\ell_{\text{CE}}$ and a variant that uses an unlikelihood term (Welleck et al., 2019) $\ell_{\text{CEUnl}}$ – all using a uniform notation for $\rho_\theta$. We use $P_m(y \mid x)$ in place of $\pi_m(y \mid x)$ to denote the probability assigned by a model $m$ to an output $y$ in a way that is agnostic to length normalization.[2] Importantly, we express each $\rho_\theta$ as a single log ratio $\log\rho_\theta^t/\rho_\theta^b$ called the **core loss equation**, which is the starting point of our analysis.

Under these core loss equations, we see that DPO variants share much structure, which is further brought out via the log ratio function $s_m(y_1, y_2)$ defined in Table 2 (using $\overline{y}$ to denote the negation of $y$, or $1 - P_m(y \mid x)$). Specifically, we see that all losses are derivable from the log ratio of winner and loser $s_\theta(y_w, y_l)$ used in SliC either exactly, as in CPO (Xu et al., 2024), or with additional log terms. DPO, for example, is expressible as this ratio minus an additional log ratio term $s_{\text{ref}}(y_w, y_l)$ that contains information about the reference model. Many variations of DPO specifically involve making the following two modifications:

**1. Adding additional terms.** Approaches like $\ell_{\text{DPOP}}$ (Pal et al., 2024) (see also Amini et al. (2024); Park et al. (2024)) incorporate additional terms into DPO ($s_{\text{ref2},\theta2}(y_w, y_w)$, see Appendix H) that address specific failure cases.

---

[1] When referring to the CPO, ORPO, and SliC losses, we refer to the losses without their original cross-entropy terms. For example, what we call SliC and ORPO refers to the cal and OR losses, respectively, in the original papers. See Appendix B for details of the original losses and our generalization.

[2] As in Zhao et al. (2025), we can define $P_m(y \mid x) := \pi_m(y \mid x)^{\frac{1}{|y|^\tau}}$ where $\tau \in \{0, 1\}$ (with $\tau = 1$ employing length norm.)

**(A) Example symbolic formulas**

Model predicts loser  Model predicts winner

$\ell_{\text{unCPO}}$

```
Implies(
    M(x,yₗ),M(x,y_w)
)
```

*Whenever the model deems the loser $(y_l)$ to be a valid generation (i.e., above $\epsilon$), it should deem the winner $(y_w)$ to be valid too.*

$\ell_{\text{CEUnl}}$

```
And(
    M(x,y_w),
    Not(M(x,yₗ)))
```

*The model should deem the winner to be valid and the loser to be not valid (i.e., below $\epsilon$).*

**(B) Model output distribution**

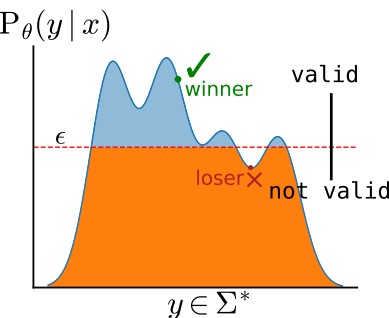

$P_\theta(y \mid x)$

✓ winner — valid

$\epsilon$

loser ✗ — not valid

$y \in \Sigma^*$

*Figure 3. What do symbolic representations of loss functions tell us?* (A) shows two symbolic formulas related to single model preference learning (involving an input $(x, y_w, y_l)$) with their semantics paraphrased in informal English. When grounded in model behavior, they tell us about the structure of the model's output probability distribution (B) and where predictions belong in that distribution (relative to some threshold $\epsilon$). We later show that these formulas correspond to $\ell_{\text{unCPO}}$ (Figure 4-5) and the common baseline $\ell_{\text{CEUnl}}$ (Table 2).

**2. Changing the reference ratio.** No reference approaches, such as $\ell_{\text{ORPO}}$ (Hong et al., 2024) and $\ell_{\text{SimPO}}$ (Meng et al., 2024), instead reparameterize the reference ratio $s_{\text{ref}}(y_w, y_l)$ either in terms of some quantity from the policy model as in ORPO ($s_\theta(\overline{y_w}, \overline{y_l})$) or a heuristic penalty term $\gamma$ as in SimPO. For SimPO we rewrite the $\gamma$ penalty term in terms of the ratio $\gamma = s_{\text{mref}}(y_w, y_l)$ (where 'mref' refers to a *manually* defined reference model simulating $\gamma$) in order to align its form with that of DPO as done by Zhao et al. (2025). For example, given any $\gamma \geq 0$, $\gamma = s_{\text{mref}}(y_w, y_l)$ can be satisfied by setting $P_{\text{mref}}(y_l \mid x) = P_{\text{mref}}(y_w \mid x)/\exp(\gamma)$ as long as the preference pairs data does not contain transitive triples or cycles.

While these approaches share a similar structure, understanding what these log ratios and extra terms in $\rho_\theta$ semantically mean remains unclear, which is the topic we discuss next. While our techniques will cover both reference and no reference approaches, due to their simplicity we use no reference losses such as $\ell_{\text{CEUnl}}$, $\ell_{\text{CPO}}$, $\ell_{\text{ORPO}}$ and a novel loss $\ell_{\text{unCPO}}$ (defined later) as running examples throughout.

## 4. Preference modeling as a reasoning problem

To better understand the DPA loss space, we will formalize the preference losses and the model quantities/log ratios $\rho_\theta$ in terms of symbolic reasoning problems. Conceptually this will involve the following core ideas and assumptions.

**Model predictions are symbolic objects.** The declarative approach involves treating LLM predictions (e.g., in $\rho_\theta$) as logical propositions. For example, when a model **M** generates an output $y_w$ for $x$, we will use $\textbf{M}(x, y_w)$ to express the logical proposition that $y_w$ is a valid generation for $x$. Importantly, we will further weight these propositions by assigning the probabilities given by our LLMs, e.g., $P_\theta(\textbf{M}(x, y_w)) = P_\theta(y_w \mid x)$. We call these our **probabilis-**

tic predictions $X_1, ..., X_n$ that underlie symbolic formulas.

**Relationships between predictions are expressed as symbolic formulas.** Relationships between model predictions take the form of symbolic constraints expressed as formulas of propositional logic P defined by applying zero or more Boolean operators over probabilistic predictions. For example, in Figure 3 (A), the top formula, which we later show is fundamental to the semantics of many DPA approaches, uses the implication operator (**Implies**) to express the constraint that model **M** should never deem the loser $y_l$ to be a valid generation ($\textbf{M}(x, y_l)$) without deeming the winner $y_w$ to also be valid ($\textbf{M}(x, y_w)$). The bottom formula tells us that only the winner $y_w$ should be deemed valid, using the conjunction and negation operators (**And**, **Not**).[3]

When grounded to model behavior via **compilation** (Section 4.1), such constraints tell us about the structure of a model's output probability distribution, as visualized in Figure 3 (B). Semantically, we assume that a valid generation is any probabilistic prediction whose weight exceeds some threshold $\epsilon$ in that distribution, similar to $\epsilon$-truncated support in Hewitt et al. (2020). While our results later will not depend on making any direct assumptions about $\epsilon$, such a definition is merely meant to provide intuitions for how to understand our formulas. For example, $\textbf{M}(x, y_l) \rightarrow \textbf{M}(x, y_w)$ dictates that whenever the loser (i.e., a point in Figure 3**B**) is found to be above the threshold $\epsilon$, the winner (another point) should also be above $\epsilon$. In other words, if the loser is deemed to be a valid generation, the winner should be too.

**Existing loss functions are expressible as symbolic formulas.** We assume that the preference loss functions in Table 2 all have an internal logic that can be expressed in the form described above. Our goal is to uncover that internal logic.

---

[3] We will switch between using conventional logical notation (e.g., $\wedge, \vee, \neg, \rightarrow, \oplus$) and operator notation (e.g., **And**, **Or**, **Not**, **Implies**, **XOR**) depending on the context.

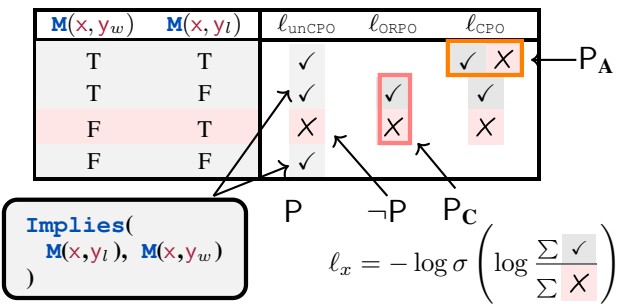

*Figure 4. Loss functions as truth tables.* The Boolean semantics (top) of our logic: ✓ s correspond to propositional models of formulas P (or $\overline{P_f}$, Eq. 4 left), ✗ s to ¬P (or $\overline{\neg P_f}$, Eq. 4 right), blank cells to conditioning constraints $P_C$ (and cells with multiple marks to $P_A$ in Eq. 4). Losses $\ell_x$ (columns) are created/recovered by assigning marks (denoting formulas) then taking weighted model counts (WMC) of ✓ and ✗ (i.e., $\sum$ ✓ and $\sum$ ✗ ) and computing the bottom equation (generalized from Eq. 3).

### 4.1. Compilation and Decompilation

Deriving the formal semantics of losses involves understanding how to robustly translate between the loss space and the symbolic space. This is reducible to the two problems we address in turn: **compilation** and **decompilation** (Fig. 1).

**Compilation and semantic loss.**  Given a symbolic formula P specifying model behavior (e.g., those in Figure 3), to compile this into a loss we interpret P in some differentiable logic. Our approach is based on probabilistic logic and the semantics of weighted model counting (WMC) (Chavira & Darwiche, 2008; Fierens et al., 2015). Accordingly, a loss for a formula is computed as the negative logarithm of the probability of that formula $p_\theta(P) = \text{WMC}(P; \theta)$ given as:

$$\text{WMC}(P; \theta) := \sum_{\mathbf{w} \models P} \prod_{\mathbf{w} \models X_i} P_\theta(X_i) \cdot \prod_{\mathbf{w} \models \neg X_i} \left(1 - P_\theta(X_i)\right).$$

This is the weighted sum over all the propositional models or truth assignments $\mathbf{w} \in \{0, 1\}^n$ of P (i.e., rows in Fig. 4) where P is satisfied (i.e., $\mathbf{w} \models P$ or rows in Fig. 4 marked with ✓ ). Each $\mathbf{w}$ is weighted via an independent product of all the probabilistic predictions $X_i$ in $\mathbf{w}$ ($P_\theta(X_i)$ or $1 - P_\theta(X_i)$ based on the truth value of $X_i$ in $\mathbf{w}$), which induces a probability distribution over all truth assignments (De Raedt & Kimmig, 2015).

Formally, the **standard semantic loss** of Xu et al. (2018) takes the form $\ell(P, \theta, D) = \mathbb{E}_{d \sim D}[- \log p_\theta(P_d)]$, where we use the notation $P_d$ throughout to refer to the substitution of variables in our formulas P (e.g., x, $y_w$, $y_l$) with specific values from $d \sim D$. Since we will later compute the probability of P conditioned (optionally) on some **conditioning constraints** $P_C$ (i.e., an additional propositional formula),

we consider the **conditional semantic loss** $\ell(P \mid P_C, \theta, D)$ and show its full objective below:

$$\min_\theta \mathbb{E}_{d \sim D} \left[ - \log p_\theta(P_d \mid P_{C_d}) \right] \qquad (2)$$

with $p_\theta(P \mid P_C) = \frac{\text{WMC}(P \wedge P_C; \theta)}{\text{WMC}(P \wedge P_C; \theta) + \text{WMC}(\neg P \wedge P_C; \theta)}$, which follows from standard conditional probability.

As an important technical point, it is easy to see that we can rewrite the formula probability (for $P \not\equiv \top$) as $p_\theta(P) = \sigma\left( \log \frac{\text{WMC}(P; \theta)}{\text{WMC}(\neg P; \theta)} \right)$, yielding a **logistic log form** of the semantic loss below that aligns with the structure of the DPA losses in Section 3. This relationship is key when translating, or decompiling, DPA losses to symbolic forms:

$$\ell(P, \theta, D) := \mathbb{E}_{d \sim D} \left[ - \log \sigma \left( \underbrace{\log \frac{\text{WMC}(P_d; \theta)}{\text{WMC}(\neg P_d; \theta)}}_{\text{sem. loss ratio } \rho_{\text{sem}}(P)} \right) \right] \quad (3)$$

As an analog to $\rho_\theta$ (Table 2), we call the inner log ratio in $\sigma(\cdot)$ above the **semantic loss ratio** of P, or $\rho_{\text{sem}}(P)$.

**Decompilation into semantic loss.**  The input in our setting is not a formula P but a particular DPA loss $\ell_x$. The goal of decompilation is to find a P that characterizes the semantics of $\ell_x$, which we treat as the inverse of compilation, i.e., P characterizes $\ell_x$ whenever its semantic loss equals $\ell_x$, that is, $\ell(P, \theta, D) = \ell_x(\theta, D)$. Given the symmetry between the DPA loss and $\rho_\theta := \log \frac{\rho_\theta^t}{\rho_\theta^b}$ (Table 2) and the semantic loss and ratio $\rho_{\text{sem}(P)} := \log \text{WMC}(P)/\text{WMC}(\neg P)$, we define **decompiling into the standard semantic loss** (Section 5.2) as translating the equations $\rho_\theta^t$ and $\rho_\theta^b$ into logical formulas $P_w$ and $P_l$ s.t. $\rho_\theta^t = \text{WMC}(P_w)$, $\rho_\theta^b = \text{WMC}(P_l)$, and there exists a single formula P where $P_w \equiv P$ and $P_l \equiv \neg P$.

We pursue this *loss equation to logic translation* approach to decompilation in Section 5.2, later using the translation rules in Table 7 for translating $\rho_\theta$ to $P_w$ and $P_l$. To make the translation direct and transparent, we impose the following **compositionality** constraint familiar from programming language semantics (Stoy, 1977):

**Assumption 1** (compositionality). *When translating the preference log ratios $\rho_\theta$ from Table 2 to propositional formulas $P_w$ and $P_l$, every unique model prediction $P_M(\cdot)$ in $\rho_\theta^t$ and $\rho_\theta^b$ is treated as a unique weighted proposition forming an atomic variable, and the propositional formulas $P_w$ and $P_l$ are built independently and compositionally by repeated application of Boolean operators over these atomic variables and none others.*

The following establishes that not all DPA losses can be compositionally decompiled using the standard semantic

loss (see proof in Appendix C involving the simplest DPA loss $\ell_{\text{CPO}}$) and motivates the need for a more expressive logic and semantic encoding of DPA, which we investigate next.

**Proposition 1** (decompilation and standard semantic loss). *Under Assumption 1, not all of the losses in Table 2 can be decompiled into the standard semantic loss (Eq 3).*

## 5. A logic for preference modeling

In the standard semantic loss, loss functions $\ell_x$ are expressible as a single propositional formulas P interpreted via probabilistic logic, with $\ell_x = -\log p_\theta(\mathsf{P})$. Proposition 1, however, reveals issues with trying to perform a compositional translation of *preference* losses into a single formula. Indeed, in logical accounts of pairwise preference (Jeffrey, 1965; Rescher, 1967), it is common to model preferences not as a single propositional formula but as an inequality between the scores $\mu$ (computed e.g., by **WMC**) of two independent propositional formulas $\mu(\mathsf{P}_w) > \mu(\mathsf{P}_l)$.

To bridge this gap, we define a **preference structure**, a relational structure and semantic encoding, that allows us to capture the semantics of DPA losses in a modular fashion using a *single* propositional formula coupled with auxiliary constraints. This structure, based on a novel construction in propositional logic (Prop. 2), makes it easy to cleanly characterize different DPA losses. We will use it to generalize the semantic loss and create a novel logic for DPA.

**Preference structure.** A preference structure is a tuple $\overline{\mathsf{P}} = (\mathsf{P}, \mathsf{P}_\mathbf{C}, \mathsf{P}_\mathbf{A})$ that, as will become clear shortly from Prop 2, captures the semantics of a winner and a loser. It consists of three propositional formulas: a **core semantic formula** P coupled with **conditioning constraints** $\mathsf{P}_\mathbf{C}$ (as in Eq 2, which restrict the propositional models that can be counted), and **additive constraints** $\mathsf{P}_\mathbf{A}$ that tell us which propositional models must always be counted. As we will show, all DPA losses in Table 2 are representable as preference structures, often ones where the same core formula P is shared (e.g., the formulas in Figure 3), differing only in their constraints ($\mathsf{P}_\mathbf{C}$ and $\mathsf{P}_\mathbf{A}$).

Each preference structure has a **formula form** $\overline{\mathsf{P}_f}$ and a **negated formula form** $\overline{\neg\mathsf{P}_f}$, defined as follows:

$$\overline{\mathsf{P}_f} := \Big(\mathsf{P} \vee \mathsf{P}_\mathbf{A}\Big) \wedge \mathsf{P}_\mathbf{C}, \quad \overline{\neg\mathsf{P}_f} := \Big(\neg\mathsf{P} \vee \mathsf{P}_\mathbf{A}\Big) \wedge \mathsf{P}_\mathbf{C}. \quad (4)$$

Intuitively, $\overline{\mathsf{P}_f}$ and $\overline{\neg\mathsf{P}_f}$ correspond to the semantics of the winner ($\mathsf{P}_w$) and the loser ($\mathsf{P}_l$). Preference structures and their corresponding formula forms are designed to give us a way to express the original semantic loss, the conditional semantic loss, and arbitrary pairwise preferences. For example, making $\mathsf{P}_\mathbf{A}$ equal to $\bot$ makes the semantic loss of $\overline{\mathsf{P}_f}$ equivalent to the conditional semantic loss from Eq 2.

*Table 3.* Different forms of the generalized semantic loss that match the structure of DPA losses in Table 1.

| Variant $\ell_{\text{sl}_x}$ | $f(\rho_{\text{sem}}, \beta) =$ | Semantic loss ratio |
|---|---|---|
| $\ell_{\text{sl-log}}$ | $-\log \sigma(\beta\rho_{\text{sem}})$ | |
| $\ell_{\text{sl-squared}}$ | $(\rho_{\text{sem}} - \frac{1}{2\beta})^2$ | $\rho_{\text{sem}}(\overline{\mathsf{P}}) := \log \frac{\text{WMC}(\overline{\mathsf{P}_f};\theta)}{\text{WMC}(\overline{\neg\mathsf{P}_f};\theta)}$ |
| $\ell_{\text{sl-margin}}$ | $\max(0, \beta - \rho_{\text{sem}})$ | |

With full preference structures containing $\mathsf{P}_\mathbf{A}$, any two propositional formulas (e.g., any $\mathsf{P}_w$ and $\mathsf{P}_l$) can be expressed as a preference structure based on a particular construction, called the **implication form**, which will play a central role when doing decompilation in Section 5.2.

**Proposition 2.** *Given any two propositional formulas $\mathsf{P}_w$ and $\mathsf{P}_l$, there exists a preference structure $\overline{\mathsf{P}} = (\mathsf{P}, \mathsf{P}_\mathbf{C}, \mathsf{P}_\mathbf{A})$ such that $\mathsf{P}_w \equiv \overline{\mathsf{P}_f}$ (Eq 4 left) and $\mathsf{P}_l \equiv \overline{\neg\mathsf{P}_f}$ (Eq 4 right).*

*Proof.* We provide a specific construction called the **implication form** of $\mathsf{P}_w$ and $\mathsf{P}_l$, based on the following logical equivalences, which can be checked manually:

$$\mathsf{P}_w \equiv \left( \underbrace{(\mathsf{P}_l \to \mathsf{P}_w)}_{\mathsf{P}} \vee \underbrace{(\mathsf{P}_w \wedge \mathsf{P}_l)}_{\mathsf{P}_\mathbf{A}} \right) \wedge \underbrace{(\mathsf{P}_w \vee \mathsf{P}_l)}_{\mathsf{P}_\mathbf{C}}$$

$$\mathsf{P}_l \equiv \left( \underbrace{\neg(\mathsf{P}_l \to \mathsf{P}_w)}_{\neg\mathsf{P}} \vee \underbrace{(\mathsf{P}_w \wedge \mathsf{P}_l)}_{\mathsf{P}_\mathbf{A}} \right) \wedge \underbrace{(\mathsf{P}_w \vee \mathsf{P}_l)}_{\mathsf{P}_\mathbf{C}}$$

As noted above, this construction corresponds exactly to the preference structure $(\mathsf{P}, \mathsf{P}_\mathbf{C}, \mathsf{P}_\mathbf{A})$ with $\mathsf{P} := \mathsf{P}_l \to \mathsf{P}_w$, $\mathsf{P}_\mathbf{C} := \mathsf{P}_w \vee \mathsf{P}_l$, and $\mathsf{P}_\mathbf{A} := \mathsf{P}_w \wedge \mathsf{P}_l$. (As a special case, when $\mathsf{P}_l \equiv \neg\mathsf{P}_w$, this simplifies to $\overline{\mathsf{P}} = (\mathsf{P}_w, \top, \bot)$.) $\square$

Figure 4 shows a natural encoding of preference structures as Boolean truth tables where rows with $\checkmark$ denote the propositional models of $\mathsf{P}_w$ and rows with $\times$ denote the propositional models of $\mathsf{P}_l$ (or, equivalently, $\overline{\mathsf{P}_f}$ and $\overline{\neg\mathsf{P}_f}$ using the implication form of $\mathsf{P}_w$ and $\mathsf{P}_l$ just introduced).

### 5.1. Semantic loss based on preference structures

In our generalization of the semantic loss, formulas P will be replaced with preference structures $\overline{\mathsf{P}}$. For example, we can modify the logistic log form of SL in Eq 3 to be $\ell(\overline{\mathsf{P}}, \theta, D)$ and change the semantic loss ratio $\rho_{\text{sem}}$ accordingly to operate over the formula forms of $\overline{\mathsf{P}}$ in Eq 4. By analogy to the generalized DPA in Eq 1, we can view this logistic log form as a particular instance of a **generalized semantic loss**:

$$\ell_{\text{sl}}(\overline{\mathsf{P}}, \theta, D) := \mathop{\mathbb{E}}_{d \sim D} \left[ f\big(\rho_{\text{sem}}(\overline{\mathsf{P}}_d), \beta\big) \right] \quad (5)$$

where, like in DPA, different choices can be made about $f$ – each giving rise to a novel loss – and the model quantities encoded in each $\overline{\mathsf{P}}$. To prove our main formal results, we

focus on the losses in Table 3, each defined with an added $\beta$ scaling parameter to match the structure of DPA losses. We note that the original semantic loss is a special case of $\ell_{\text{sl-log}}$ (specifically in cases where the input preference structures have $\mathsf{P_C} \equiv \top$ and $\mathsf{P_A} \equiv \bot$).

**How many loss functions are there?** Under this formulation, we can view loss creation as a generative procedure: select an $f$ then sample two formulas $\mathsf{P}_w$ and $\mathsf{P}_l$ (each denoting a unique Boolean function in $n$ variables) to create a $\overline{\mathsf{P}}$ via Prop 2 (see also Figure 4). Absent any constraints, the total number of definable preference structures is doubly exponential in the number of probabilistic predictions $n$, specifically $4^{2^n}$ (i.e., all unique pairs of Boolean functions). While not all such preference structures will lead to meaningful or unique losses, for DPO ($n = 4$), this results in an upper bound of about 4.3 billion definable losses.

**How is the loss space structured?** While the space is large, one can structure this space using the semantics of the corresponding formulas. Below we define preference structure *entailment* and *equivalence*, and relate these semantic notions to the behavior of the compiled losses. These formal notions not only give us tools for structuring the DPA loss space but also inform the search for new loss functions.

We define **preference entailment** for two preference structures $\overline{\mathsf{P}}^{(1)} \sqsubseteq \overline{\mathsf{P}}^{(2)}$ in terms of ordinary propositional entailment ($\models$) between their formula forms: $\overline{\mathsf{P}}^{(1)} \sqsubseteq \overline{\mathsf{P}}^{(2)} :=$ $(\overline{\mathsf{P}_f}^{(1)} \models \overline{\mathsf{P}_f}^{(2)} \ \wedge \ \overline{\neg\mathsf{P}_f}^{(2)} \models \overline{\neg\mathsf{P}_f}^{(1)})$. These losses are monototic w.r.t. preference entailment (proof deferred to Appendix E), as in the original SL (Xu et al., 2018):

**Proposition 3** (monotonicity). *If* $\overline{\mathsf{P}}^{(1)} \sqsubseteq \overline{\mathsf{P}}^{(2)}$ *then* $\ell_{sl}(\overline{\mathsf{P}}^{(1)}, \theta, D) \geq \ell_{sl}(\overline{\mathsf{P}}^{(2)}, \theta, D)$ *for any* $\theta, D$.

We will later use entailment to characterize the relative strength of DPA losses and visualize their relations using a representation called a **loss lattice** (see Figure 5). We also extend entailment to **preference equivalence** $\overline{\mathsf{P}}^{(1)} \equiv \overline{\mathsf{P}}^{(2)}$ in the natural way, namely when $\overline{\mathsf{P}}^{(1)} \sqsubseteq \overline{\mathsf{P}}^{(2)}$ and $\overline{\mathsf{P}}^{(2)} \sqsubseteq \overline{\mathsf{P}}^{(1)}$. Equivalent preference structures have identical semantic losses (see Corollary 1 in Appendix E).

### 5.2. Decompiling DPA losses into preference structures

The **decompilation** of a DPA loss $\ell_{\text{DPA}_x}$ into a symbolic form can now be stated as finding a preference structure $\overline{\mathsf{P}}$ whose particular semantic loss $\ell_{\text{sl}_x}$ is equal to $\ell_{\text{DPA}_x}$:

$$\ell_{\text{DPA}_x}(\theta, D) = \ell_{\text{sl}_x}(\overline{\mathsf{P}}, \theta, D) \qquad (6)$$

**Correct characterization.** We say that a preference structure $\overline{\mathsf{P}}$ **correctly characterizes** a loss $\ell_x$ under a particular

---

**Algorithm 1** Translation of loss to logic (**decompilation**)

**input** Disjoint polynomial $\rho_\theta = \log \frac{\rho_\theta^t}{\rho_\theta^b}$  **output** $\overline{\mathsf{P}}$

$\mathsf{P}_t \leftarrow \text{SEM}(\rho_\theta^t)$   {Translation to logic, Table 7}
$\mathsf{P}_b \leftarrow \text{SEM}(\rho_\theta^b)$
$\mathsf{P} \leftarrow \text{SIMPLIFY}(\textbf{Implies}(\mathsf{P}_b, \mathsf{P}_t))$ {Implication form}
$\mathsf{P_C} \leftarrow \text{SIMPLIFY}(\textbf{Or}(\mathsf{P}_t, \mathsf{P}_b))$      {via Proposition 2}
$\mathsf{P_A} \leftarrow \text{SIMPLIFY}(\textbf{And}(\mathsf{P}_t, \mathsf{P}_b))$

**return** $\overline{\mathsf{P}} := (\mathsf{P}, \mathsf{P_C}, \mathsf{P_A})$   $\{\rho_\theta = \log \frac{\text{WMC}(\overline{\mathsf{P}_f};\theta)}{\text{WMC}(\overline{\neg\mathsf{P}_f};\theta)}$, Lem. 1$\}$

---

$\ell_{\text{sl}_x}$ whenever the condition in Eq 6 holds. Given the structure of the DPA loss (Eq 1) and the generalized semantic loss (Eq. 5), for any fixed $f$, this can be reduced to finding a $\overline{\mathsf{P}}$ whose semantic loss ratio $\rho_{\text{sem}}(\overline{\mathsf{P}})$ is equal to $\ell_x$'s core loss equation $\rho_\theta$, i.e.:

$$\log \frac{\rho_\theta^t}{\rho_\theta^b} = \log \frac{\text{WMC}(\overline{\mathsf{P}_f};\theta)}{\text{WMC}(\overline{\neg\mathsf{P}_f};\theta)} \qquad (7)$$

Based on this, we define a procedure for translating the core loss equations $\rho_\theta$ in Table 2 into preference structures.

**Characterizing the DPA equation class.** Motivated by the structure of real preference losses, we assume that by construction all the core equations for DPA losses $\rho_\theta^t$ and $\rho_\theta^b$ are expressible as certain types of **disjoint multilinear polynomials** over binary variables $\{x_i\}_{i=1}^n$, intuitively polynomials whose translation via the rules in Table 7 results in valid formulas of propositional logic. Formally, such polynomials over $n$ variables are defined as any polynomial $e$ of the form $e = \sum_i e_i$ where (a) for all $i$ there exists $J_i \subseteq \{1, \ldots, n\}$ such that $e_i = \prod_{j \in J_i} \ell_{ij}$ where $\ell_{ij}$ is either $x_j$ or $(1 - x_j)$, and (b) for all $i, i'$, terms $e_i$ and $e_{i'}$ are disjoint, i.e., have no common solutions (for some $k$, one term has $x_k$ and the other has $1 - x_k$) (see Appendix H for discussion of a loss that doesn't immediately fit this format).

**Translation algorithm.** Our translation process is shown in Algorithm 1 along with an example in Figure 2. Given $\rho_\theta$, both $\rho_\theta^t$ and $\rho_\theta^b$ are independently translated into logic via a compositional translation function SEM. The translation is standard, based on the rules in Table 7: first each model prediction $\mathsf{P_M}(\cdot)$ is mapped to a probabilistic prediction vaiable $\mathbf{M}(\cdot)$; then $1 - \mathsf{P}$ is mapped to negation, $\mathsf{P}_1 \cdot \mathsf{P}_2$ to conjunction, and $\mathsf{P}_1 + \mathsf{P}_2$ to disjunction; these rules are applied repeatedly until the full expression is translated. By induction on the rules, one can establish the correctness of the translation function SEM, i.e., that for any disjoint multilinear polynomial $\rho_\theta^z$, it holds that $\rho_\theta^z = \text{WMC}(\text{SEM}(\rho_\theta^z); \theta)$. Finally, the implication construction from Prop 2 is applied to create a preference structure $\overline{\mathsf{P}}$, where formulas are (optionally) minimized via SIMPLIFY (see example in Fig. 9).

The following follows from the correctness of our translation

*Table 4. What do formalized versions of standard losses look like?* Formalizations of some of the losses from Table 2 shown in terms of P (their core semantic formula) and conditioning constraints $P_C$ (for succinctness, we exclude $P_A$, which can be inferred from each $P_C$ via Algorithm 1).

| Loss | Representation $\overline{P}$ |
|---|---|
| CE | $P := \mathbf{M}(x, y_w)$, $P_C := \bot$ |
| CEUnl | $P := \mathbf{And}(\mathbf{M}(x, y_w), \mathbf{Not}(\mathbf{M}(x, y_l)))$
$P_C := \bot$ |
| CPO | $P := \mathbf{Implies}(\mathbf{M}(x, y_l), \mathbf{M}(x, y_w))$
$P_C := \mathbf{Or}(\mathbf{M}(x, y_l), \mathbf{M}(x, y_w))$ |
| ORPO | $P := \mathbf{Implies}(\mathbf{M}(x, y_l), \mathbf{M}(x, y_w))$
$P_C := \mathbf{XOR}(\mathbf{M}(x, y_l), \mathbf{M}(x, y_w))$ |
| DPO | $P := \mathbf{Implies}($
$\quad \mathbf{And}(\mathbf{Ref}(x, y_w), \mathbf{M}(x, y_l)),$
$\quad \mathbf{And}(\mathbf{Ref}(x, y_l), \mathbf{M}(x, y_w)))$
$P_C := \mathbf{Or}(\mathbf{And}(\mathbf{Ref}(x, y_w), \mathbf{M}(x, y_l)),$
$\quad \mathbf{And}(\mathbf{Ref}(x, y_l), \mathbf{M}(x, y_w)))$ |
| SimPO | $P := \mathbf{Implies}($
$\quad \mathbf{And}(\mathbf{Mref}(x, y_w), \mathbf{M}(x, y_l)),$
$\quad \mathbf{And}(\mathbf{Mref}(x, y_l), \mathbf{M}(x, y_w)))$
$P_C := \mathbf{Or}(\mathbf{And}(\mathbf{Mref}(x, y_w), \mathbf{M}(x, y_l)),$
$\quad \mathbf{And}(\mathbf{Mref}(x, y_l), \mathbf{M}(x, y_w)))$ |

rules and the implication construction (Prop 2):

**Lemma 1** (correctness of translation). *Given a loss equation $\rho_\theta := \log \rho_\theta^t / \rho_\theta^b$ with disjoint multilinear polynomials $\rho_\theta^t$, and $\rho_\theta^b$, Algorithm 1 returns a preference structure $\overline{P}$ whose semantic loss ratio $\rho_{sem}(\overline{P})$ equals $\rho_\theta$.*

This establishes the correctness of our decompilation algorithm, showing specifically that Algorithm 1 yields preference structures that satisfy Eq 7.

# 6. Results and Discussion

Table 4 shows the preference structures obtained from Algorithm 1 for the DPA losses in Table 2. The following result establishes their correctness:

**Theorem 1** (Correctness of formalization). *The preference structures in Table 4 correctly characterize the losses in Table 2 and satisfy Eq 6 under semantic loss $\ell_{sl\text{-}log}$ (Table 3).*

*Proof.* Since the original losses were all formulated using the logistic log form of DPA, the correctness of Algorithm 1 (which follows from Lemma 1) implies that compiling the representations in Table 4 (which, as noted above, were obtained by running Algorithm 1 on the losses in Table 2) under $\ell_{sl\text{-}log}$ will yield precisely the original losses, and hence satisfies Eq 6. $\qquad \square$

By changing the version of semantic loss, we can extend our analysis to other variants of DPO, showing the generality of our semantic analysis and its invariance to the choice

of $f$. For example, by changing $\ell_{sl\text{-}log}$ to $\ell_{sl\text{-}squared}$ or $\ell_{sl\text{-}margin}$, we immediately obtain the following results:

**Theorem 2** (Extension to other DPO). *The DPO and CPO preference structures in Table 4 correctly characterize the IPO and SliC losses (Table 1) and satisfy Eq 6 under the $\ell_{sl\text{-}squared}$ and $\ell_{sl\text{-}margin}$ semantic losses, respectively.*

Given the ubiquity of DPO-style updates in other *online* variants of DPA (Qi et al., 2024; Zhang et al., 2024; Chen et al., 2024b; Guo et al., 2024), similar formal characterizations could be extended to these variants, as well as to recent reward distillation approaches such as Fisch et al. (2024), which we see as a promising future direction of research.

## 6.1. What do we learn about known losses?

**Single model approaches have an intuitive semantics, but differ in conditioning constraints.** One goal of our formalization is to cleanly characterize the semantic relationships between losses. For example, with CPO and ORPO we see that both are derivable from the same core semantic formula $P := \mathbf{M}(x, y_l) \rightarrow \mathbf{M}(x, y_w)$ from Figure 3. Indeed, this formula appears to capture the fundamental semantics of many known DPA losses. Under our account, however, losses differ in the conditioning constraints $P_C$ they impose, which structure the underlying probability distribution in different ways. CPO conditions P on a **one-true** constraint that requires *at least one* of the winner and the loser to be deemed valid (i.e., rules out the semantic interpretation where both are deemed invalid; last row in Fig. 4 top), whereas ORPO imposes a **one-hot** constraint where exactly one must be deemed valid. Through further semantic analysis of their preference structures, we can see that both losses are semantically entailed by $\ell_{CEUnl}$, yet have a non-entailing relation to one another or to the cross-entropy loss $\ell_{CE}$.

Interestingly, we see that preference losses are highly constrained, which might explain their success. This is in contrast to the losses typically compiled via semantic loss in neuro-symbolic modeling (Marra et al., 2024), suggesting that there is much to learn by working backward from empirically successful loss functions to their semantic properties.

**There are many single model losses still to explore, and we can exhaustively enumerate them.** Another goal of ours is to be able to derive novel losses from first principles. This can be achieved by modifying the conditioning constraints $P_C$. Figure 5 shows a (non-exhaustive) lattice representation of the loss landscape for single model preference approaches (see Fig. 8 for an exhaustive variant) created by mechanically manipulating the constraints in $\ell_{CEUnl}$ (the most constrained) and ordering the resulting losses by strict entailment ($\sqsubset$), terminating in $\ell_{unCPO}$ (our constraint-free running example from Figs. 3-4).

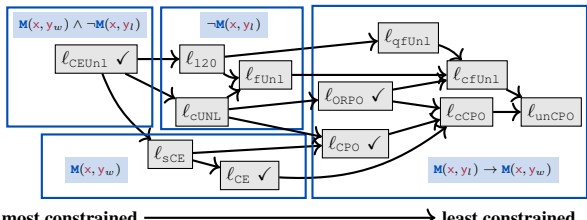

*Figure 5. What other losses are there?* Here we show the loss landscape for single model preference approaches using a **loss lattice** showing losses (nodes) structured according to strict entailment ($\sqsubseteq$) and their core formulas $\boxed{\text{P}}$ (boxes) with ✓ being the known losses. See Appendix F for details of the individual losses and a more exhaustive lattice with DPO variants in Figure 8.

We structure the resulting set of losses using the core loss equations P, which reveals different **semantic regions** (blue boxes). In addition to novel variants of $\ell_{\text{CPO}}$ and $\ell_{\text{ORPO}}$ that optimize for $\mathbf{M}(\mathsf{x}, \mathsf{y}_l) \to \mathbf{M}(\mathsf{x}, \mathsf{y}_w)$, we see an entirely unexplored region of unlikelihood losses ($\ell_{\text{l2O}}, \ell_{\text{cUNL}}, \ell_{\text{fUnl}}$) that optimize for the negation of the loser $\neg\mathbf{M}(\mathsf{x}, \mathsf{y}_l)$. Through compilation, all new losses can be explored experimentally, which we discuss below.

**Adding a reference model has a clear, though sometimes peculiar, semantics.** The semantics of DPO, is shown in Table 4 and is logically equivalent to a conjunction of two implications: $\mathbf{Ref}(\mathsf{x}, \mathsf{y}_w) \wedge \mathbf{M}(\mathsf{x}, \mathsf{y}_l) \to \mathbf{M}(\mathsf{x}, \mathsf{y}_w)$ and $\mathbf{Ref}(\mathsf{x}, \mathsf{y}_w) \wedge \neg\mathbf{Ref}(\mathsf{x}, \mathsf{y}_l) \to \neg\mathbf{M}(\mathsf{x}, \mathsf{y}_l)$. The first says that *If the reference deems the winner to be valid and the tunable model deems the loser to be valid, then that model should also deem the winner to be valid*, while the second says that *the tunable model should deem the loser to be not valid whenever the reference deems the winner to be valid and the loser to be not valid*. While this semantics makes sense, and complements nicely the semantics of CPO by adding information about the referent model, DPO includes conditioning constraints that are hard to justify from first principles, and that make it semantically disconnected from the CE and CEUnl baselines.

We also note that variants like SimPO and DPOP when formalized maintain exactly the same structure of DPO in Table 4, with DPOP adding repeated variables that amplify the score of the winner (see Appendix H). Giving the semantic similarity between these variants and DPO, any small semantic change found in one would likely be useful in these others, which motivates general exploration into varying the conditioning constraints. Several such variants of DPO and SimPO are shown in Figure 8.

**Can we find empirically improved losses using our method?** The ultimate goal of our analysis is to facilitate the discovery of empirically improved DPA losses. As

a case study, we implemented single model losses around the known $\ell_{\text{CPO}}$ in Figure 5, treating it as a baseline to improve upon. Using a model-as-judge style evaluation adapted from Hong et al. (2024) and a Qwen-0.5B LLM (details in Appendix G), we found one particular loss, $\ell_{\text{cCPO}}$ to be competitive with $\ell_{\text{CPO}}$, achieving a win-rate of $52.0$ as shown in Table 5. We also observe that different losses have markedly different performance across different datasets subsets, suggesting that a one-size-fits-all approach isn't ideal—semantically different tasks are best learned using different losses.

*Table 5.* Results of a feasibility study involving Qwen-0.5B tuned on the new losses (rows) compared against the known loss $\ell_{\text{CPO}}$ (second column) on ultrafeedback (Cui et al., 2024) test in aggregate (2nd column) and on subsets (right columns). See details in Section G.

| loss | WR% ($\ell_{\text{cpo}}$) | evol | false-qa | flan | sharegpt | ultrachat |
|---|---|---|---|---|---|---|
| $\ell_{\text{cfUNL}}$ | 46.1 ($\pm$0.4) | 46.1 ($\pm$2.2) | 51.6 ($\pm$2.9) | 46.4 ($\pm$1.7) | 46.2 ($\pm$1.2) | 44.1 ($\pm$1.0) |
| $\ell_{\text{qfUNL}}$ | 48.9 ($\pm$0.8) | 45.3 ($\pm$1.9) | 34.7 ($\pm$6.3) | 57.9 ($\pm$1.2) | 46.8 ($\pm$2.4) | 41.3 ($\pm$1.4) |
| $\ell_{\text{cCPO}}$ | 52.0 ($\pm$0.6) | 50.7 ($\pm$0.5) | 50.2 ($\pm$0.7) | 57.2 ($\pm$1.1) | 47.2 ($\pm$1.8) | 53.1 ($\pm$1.9) |
| $\ell_{\text{unCPO}}$ | 46.0 ($\pm$0.2) | 45.8 ($\pm$0.3) | 52.1 ($\pm$3.0) | 45.7 ($\pm$0.6) | 46.2 ($\pm$2.1) | 44.8 ($\pm$2.1) |

While small scale, this study demonstrates the feasibility of using our framework to derive empirically successful losses. Appendix G reports additional experiments and findings. We include further details about the log probability behavior of different losses and how such behavior relates to the *constrainedness* of each loss. We see further work on empirically exploring these new losses on a wider range of models and scales as a promising direction of future research, as well as using the full power of logic to derive even more complex losses from first principles.

## 7. Conclusion

Despite the routine use of a variety of DPA algorithms to align LLMs with human preferences, knowing what exactly the losses underlying these algorithms capture and how they relate to each other remains largely unknown. We presented a new technique for characterizing the semantics of such losses in terms of logical formulas over boolean propositions that capture model predictions. Key to our approach is a *decompilation* procedure, allowing one to compositionally derive provably correct symbolic formulas corresponding to any loss function expressed as a ratio of disjoint multilinear polynomials. Our approach provides a fresh perspective into preference losses, identifying a rich loss landscape and opening up new ways for practitioners to explore new losses by systematically varying the symbolic formulas corresponding to existing successful loss functions.

## Acknowledgements

We thank the following people, as well as the UtahNLP group and anonymous reviewers, for their feedback at vari-

ous stages of the work: Kareem Ahmed, Gregor Betz, Junyan Cheng, Hamish Ivison, Maryna Kavalenka, Emile van Krieken, Nathan Lambert, Robin Manhaeve, Valentina Pyatkin, Antonio Vergari, and Gijs Wijnholds. VS was supported in part by NSF awards #2217154 and #2411319.

## Impact Statement

This paper presents work whose goal is to advance the field of Machine Learning. There are many potential societal consequences of our work, none which we feel must be specifically highlighted here.

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

## A. Additional related work

**Decompilation of losses.** In contrast to work on loss compilation, we focus on the inverse problem of **decompilation** (see Friedman et al. (2024)), or deriving symbolic expressions from known loss functions. Work in this area has mostly been limited to symbolically deriving standard loss function such as cross-entropy (Giannini et al., 2020; Li et al., 2019), whereas we look at deriving the semantics of more complex LLM algorithms.

**Declarative model programming.** In addition to neuro-symbolic modeling (Richardson & Srikumar, 2023; Marra et al., 2024), we take inspiration from recent work on formalizing LLM algorithms in terms of programming language concepts (Dohan et al., 2022; Beurer-Kellner et al., 2023; Khattab et al., 2024), with our approach being declarative in style (see review in Richardson & Wijnholds (2024)). As such, our study takes much inspiration from the large literature on declarative programming techniques for ML (Eisner et al., 2004; De Raedt et al., 2007; Li et al., 2023; Vieira et al., 2017; Ślusarz et al., 2023; van Krieken et al., 2024a; Hinnerichs et al., 2024).

## B. Original losses

Further details of the original losses in Table 2, along with other variants such as R-DPO (Park et al., 2024), ODPO (Amini et al., 2024), and DPKD (Yixing et al., 2024), are shown in Table 6. While our formalization abstracts over certain details such as length normalization and additional regularization terms (and hence generalizes/simplifies some of the original losses), we include here such details from the original studies. In the case of regularization terms, as noted in Table 6, most **no reference** approaches add an additional cross-entropy term, often making the full losses in these studies equal to $\ell_{x+\text{CE}} = \ell_x + \lambda \ell_{\text{CE}}$ (with weight $\lambda$). In some cases, additional terms $\alpha$ are assumed that we abstract over in our analysis, e.g., in $\ell_{\text{ORPO}}$ the full loss includes an additional weight term $\alpha$ that is added to the main loss (in our experiments below, $\alpha$ is implicitly set to 1).

## C. Compositionality constraint

**Proposition 1** (decompilation and standard semantic loss)**.** *Under Assumption 1, not all of the losses in Table 2 can be decompiled into the standard semantic loss (Eq 3).*

*Proof.* Taking $\ell_{\text{CPO}}$ as an example, the loss equation is based on the ratio $s_\theta(y_w, y_l)$ consisting of two predictions $P_\theta(y_w \mid x)$ and $P_\theta(y_l \mid x)$, which we can translate into the propositional formulas $\mathsf{P}_t := \mathbf{M}(\mathsf{x}, \mathsf{y}_w)$ and $\mathsf{P}_b := \mathbf{M}(\mathsf{x}, \mathsf{y}_l)$, consisting of a total of two atomic propositions. Translating this to the standard semantic loss involves finding a *single* $\mathsf{P}$ such that $\mathsf{P}_w = \mathsf{P}$ and $\mathsf{P}_l = \neg\mathsf{P}$. To see that no such $\mathsf{P}$ exists, we can enumerate all 16 unique Boolean functions over variables $\mathbf{M}(\mathsf{x}, \mathsf{y}_w)$ and $\mathbf{M}(\mathsf{x}, \mathsf{y}_l)$ (the only variables we are allowed under Assumption 1) and verify that none yield a single formula $\mathsf{P}$ s.t. $\log \frac{\text{WMC}(\mathsf{P};\theta)}{\text{WMC}(\neg\mathsf{P};\theta)} = s_\theta(y_w, y_l)$. The same argument can be applied to each of the other non-baseline losses in the table. $\qquad\square$

Without the compositionality assumption, one can encode any $\rho_\theta$ as a formula using additional variables and weighting schemes, as is commonly done in standard WMC encodings (Chavira & Darwiche, 2008). However, the semantics of the resulting formulas are less transparent and often hidden in the weights. We instead propose to define below a novel (unweighted) encoding for preference that doesn't require additional variables, thereby facilitating a compositional and transparent translation from loss equations.

## D. Semantic translation rules

In Table 7 we show the full translation rules for Algorithm 1.

*Table 6.* Details of the original losses from Table 2 and others (adapted from Meng et al. (2024)), all of which were originally implemented using the logistic log-loss, i.e., each $\ell_x = -\log \sigma(\beta \rho_\theta)$. We also include details about whether cross-entropy regularization (**CE term**) and length normalization (**length norm.**) were used (yes ✓, no ✗) along with other details (**Extra details**) (e.g., extra weight terms, specific choices about $\beta$ or cross-entropy weight $\lambda$) that we either exclude or generalize in our analysis and experiments (e.g., extra loss weighting terms $\alpha$). See Winata et al. (2025) for a comprehensive review and Zhao et al. (2025) for an approach that further mixes the DPO and SimPO losses.

| Loss name | core loss equation $\rho_\theta$ | CE term | length norm. | Extra details and terms |
|---|---|---|---|---|
| | | **common baseline losses** | | |
| $\ell_{\text{CE}}$ | $\log \frac{P_\theta(y_w\|x)}{1-P_\theta(y_w\|x)}$ | – | – | |
| $\ell_{\text{CEUnl}}$ (Rafailov et al., 2023) | $\log \frac{P_\theta(y_w\|x)(1-P_\theta(y_l\|x))}{1-(P_\theta(y_w\|x)(1-P_\theta(y_l\|x)))}$ | – | – | Unlikelihood term weighted by $\alpha$ |
| | | **reference approaches** | | |
| $\ell_{\text{DPO}}$ (Rafailov et al., 2023) | $\log \frac{P_\theta(y_w\|x)P_{\text{ref}}(y_l\|x)}{P_{\text{ref}}(y_w\|x)P_\theta(y_l\|x)}$ | ✗ | ✗ | |
| $\ell_{\text{ODPO}}$ (Amini et al., 2024) | $\log \frac{P_\theta(y_w\|x)P_{\text{ref}}(y_l\|x)}{P_{\text{ref}}(y_w\|x)P_\theta(y_l\|x)} - \gamma_{\text{offset}}$ | ✗ | ✗ | Added offset term $\gamma_{\text{offset}}$ |
| $\ell_{\text{DPOP}}$ (Pal et al., 2024) | $\log \frac{P_\theta(y_w\|x)P_{\theta2}(y_w\|x)P_{\text{ref}}(y_l\|x)}{P_{\text{ref}}(y_w\|x)P_{\text{ref2}}(y_w\|x)P_\theta(y_l\|x)}$ | ✗ | ✗ | See Appendix H |
| $\ell_{\text{R-DPO}}$ (Park et al., 2024) | $\log \frac{P_\theta(y_w\|x)P_{\text{ref}}(y_l\|x)}{P_{\text{ref}}(y_w\|x)P_\theta(y_l\|x)} + \gamma_{\text{len}}$ | ✗ | ✗ | Added length bias term $\gamma_{\text{len}}$ |
| $\ell_{\text{DPKD}}$ (Yixing et al., 2024) | $\log \frac{P_{\text{student}}(y_w\|x)P_{\text{teacher}}(y_l\|x)}{P_{\text{teacher}}(y_w\|x)P_{\text{student}}(y_l\|x)}$ | ✓ | ✓ | Distillation, re-parameterizes `ref` and $\theta$ |
| | | **single model (no reference)**, CE weight $\lambda$ | | |
| $\ell_{\text{CPO}}$ (Xu et al., 2024) | $\log \frac{P_\theta(y_w\|x)}{P_\theta(y_l\|x)}$ | ✓ | ✗ | Removes `ref` |
| $\ell_{\text{ORPO}}$ (Hong et al., 2024) | $\log \frac{P_\theta(y_w\|x)(1-P_\theta(y_l\|x))}{P_\theta(y_l\|x)(1-P_\theta(y_w\|x))}$ | ✓ | ✓ | $\beta=1$, main loss weighted by $\alpha$, $\lambda=1$ |
| $\ell_{\text{SimPO}}$ (Meng et al., 2024) | $\log \frac{P_\theta(y_w\|x)}{P_\theta(y_l\|x)} - \gamma$ | ✗ | ✓ | Added margin term $\gamma$, re-formalized in Table 2 |

*Table 7.* Rules for the compositional translation of loss expressions into symbolic formulas. See again example in Figure 2.

| Input | SEM($\cdot$) |
|---|---|
| predictions | |
| $P_{\mathbf{M}}(\mathsf{y} \mid \mathsf{x})$ | $\mathsf{P} := \mathbf{M}(\mathsf{x}, \mathsf{y})$ |
| formulas $\mathsf{P}$ | |
| $\mathsf{P}_1 \cdot \mathsf{P}_2$ | $\mathsf{P} := \mathbf{And}(\mathsf{P}_1, \mathsf{P}_2)$ |
| $1 - \mathsf{P}$ | $\mathsf{P} := \mathbf{Not}(\mathsf{P})$ |
| $\mathsf{P}_1 + \mathsf{P}_2$ | $\mathsf{P} := \mathbf{Or}(\mathsf{P}_1, \mathsf{P}_2)$ |

## E. Proofs of other propositions

Below we state propositions discussed in Section 5.1 with their proofs.

**Proposition 3** (monotonicity). *If* $\overline{\mathsf{P}}^{(1)} \sqsubseteq \overline{\mathsf{P}}^{(2)}$ *then* $\ell_{sl}(\overline{\mathsf{P}}^{(1)}, \theta, D) \geq \ell_{sl}(\overline{\mathsf{P}}^{(2)}, \theta, D)$ *for any* $\theta, D$.

*Proof.* By the definition of preference entailment, we have $\overline{\mathsf{P}}_f^{(1)} \models \overline{\mathsf{P}}_f^{(2)}$. This means that for any $d$, $\overline{\mathsf{P}}_f^1(d) \models \overline{\mathsf{P}}_f^2(d)$, which implies that for any $\theta$, $\text{WMC}(\overline{\mathsf{P}}^{(1)}(d); \theta) \leq \text{WMC}(\overline{\mathsf{P}}^{(2)}(d); \theta)$. From the definition of preference entailment, we also have $\overline{\neg \mathsf{P}}^{(2)}(d) \models \overline{\neg \mathsf{P}}^{(1)}(d)$. Following a similar line of reasoning as above, this implies $\text{WMC}(\overline{\neg \mathsf{P}}^{(1)}(d); \theta) \geq \text{WMC}(\overline{\neg \mathsf{P}}^{(2)}(d); \theta)$. Thus, for any $d$ and $\theta$, the weighted model counting ratio term in the semantic loss in Table 3 is no larger for $\overline{\mathsf{P}}^{(1)}$ than for $\overline{\mathsf{P}}^{(2)}$.

It follows that $\ell_{\text{sl}}(\overline{\mathsf{P}}^{(1)}, \theta, \{d\}) \geq \ell_{\text{sl}}(\overline{\mathsf{P}}^{(2)}, \theta, \{d\})$. Taking the expectation over $d \sim D$, we obtain $\ell_{\text{sl}}(\overline{\mathsf{P}}^{(1)}, \theta, D) \geq \ell_{\text{sl}}(\overline{\mathsf{P}}^{(2)}, \theta, D)$. $\qquad \square$

It follows that equivalent preference structures have identical semantic losses:

**Corollary 1** (semantic equivalence). *If* $\overline{\mathsf{P}}^1 \equiv \overline{\mathsf{P}}^2$ *then* $\ell_{sl}(\overline{\mathsf{P}}^{(1)}, \theta, D) = \ell_{sl}(\overline{\mathsf{P}}^2, \theta, D)$ *for any* $\theta, D$.

The next result is an analogue to the locality property in the original semantic loss (Xu et al., 2018), which tells us that unused logical variables in formulas do not affect loss values, which allows us to compare losses with different number of variables.

**Proposition 4** (locality). *Let* $\overline{\mathsf{P}}$ *be a preference structure defined over probabilistic prediction variables* $\mathbf{X}$ *with parameters* $\theta_x$. *Let* $\mathbf{Y}$ *be some disjoint set of variables with parameters* $\theta_y$. *Then* $\ell_{sl}(\overline{\mathsf{P}}, \theta_x, D) = \ell_{sl}(\overline{\mathsf{P}}, [\theta_x \theta_y], D)$ *for any* $D$.

*Proof.* Let $\mathbf{w}_x$ be any world over variables $\mathbf{X}$ and $\mathbf{w}_y$ be any world over (disjoint) variables $\mathbf{Y}$. Let $\mathbf{w}_{x,y}$ denote the joint world. By the standard semantic loss, the probability of the world $\mathbf{w}_{x,y}$ in the $(\mathbf{X}, \mathbf{Y})$ space can be written as $P_{\theta_x, \theta_y}(\mathbf{w}_{x,y}) = \prod_{X_i \in \mathbf{X}} Q_{\theta_x, \theta_y}(X_i) \cdot \prod_{Y_j \in \mathbf{Y}} Q_{\theta_x, \theta_y}(Y_j)$ where $Q$ is either $P$ or $1 - P$. Since the parameters $\theta_x$ and $\theta_y$ refer to disjoint sets of variables, we can simplify this to $\prod_{X_i \in \mathbf{X}} Q_{\theta_x}(X_i) \cdot \prod_{Y_j \in \mathbf{Y}} Q_{\theta_y}(Y_j)$.

It follows that the marginal probability of the world

$\mathbf{w}_x$ in the $(\mathbf{X}, \mathbf{Y})$ space equals $P_{\theta_x, \theta_y}(\mathbf{w}_x) = \sum_{\mathbf{Y}} \left( \prod_{X_i \in \mathbf{X}} Q_{\theta_x}(X_i) \cdot \prod_{Y_j \in \mathbf{Y}} Q_{\theta_y}(Y_j) \right) = \prod_{X_i \in \mathbf{X}} Q_{\theta_x}(X_i) \cdot \sum_{\mathbf{Y}} \left( \prod_{Y_j \in \mathbf{Y}} Q_{\theta_y}(Y_j) \right) = \prod_{X_i \in \mathbf{X}} Q_{\theta_x}(X_i) \cdot \prod_{Y_j \in \mathbf{Y}} \left( Q_{\theta_y}(Y_j) + (1 - Q_{\theta_y}(Y_j)) \right) = \prod_{X_i \in \mathbf{X}} Q_{\theta_x}(X_i) = P_{\theta_x}(\mathbf{w}_x)$. This last expression is precisely the probability of the world $\mathbf{w}_x$ in only the $\mathbf{X}$ space. Thus, $P_{\theta_x}(\mathbf{w}_x) = P_{\theta_x, \theta_y}(\mathbf{w}_x)$, which implies $\mathrm{WMC}(\overline{\mathsf{P}}; \theta_x) = \mathrm{WMC}(\overline{\mathsf{P}}; \theta_x, \theta_y)$ and similarly for $\overline{\neg \mathsf{P}}$. From this, the claim follows immediately. $\qquad \square$

## F. New losses in loss lattice

To visualize the semantics of the single model losses shown in Figure 5, we use the Boolean truth table shown in Figure 6. As already illustrated in Figure 4, each loss column can be mechanically converted into a preference structure via the following steps: 1) translate $\checkmark$ and $\times$ into two standard propositional formulas that are logically consistent with the marks, $\mathsf{P}_t$ for $\mathsf{P}_b$, respectively, then 2) apply the rules in Algorithm 1 to these formulas to get a preference structure $\overline{\mathsf{P}}$. (Note that the formulas in boxes in Figure 5 show the core formula $\mathsf{P}$ in the resulting preference structure and intentionally hide details about the constraints.)

With these preference structures, we can then obtain a compiled version of the loss by simply applying one of the versions of the semantic loss. In simplified terms, finding the compiled loss equation directly from a truth table for a given version of semantic loss with convex function $f$ (e.g., those listed in Table 3) involves the following

$$ f\left( \log \frac{\sum \boxed{\checkmark}}{\sum \boxed{\times}} \right) $$

where we can replace each $\sum .$ with the corresponding WMC equations for each mark, then simplify the resulting equation (i.e., the core loss equation) to arrive at a compact loss equation that can be directly used for implementation.

**Losses used in experiments** Employing the process above, below we show the core loss equations for the losses we used in our experiments in accordance with the form in Table 2:

| Loss name | Core loss equation (implementation) |
|---|---|
| $\ell_{\mathrm{cpo}}$ | $\log \frac{P_\theta(y_w \mid x)}{P_\theta(y_l \mid x)}$ |
| $\ell_{\mathrm{orpo}}$ | $\log \frac{P_\theta(y_w \mid x)(1 - p_\theta(y_l \mid x))}{P_\theta(y_l \mid x)(1 - p_\theta(y_w \mid x))}$ |
| $\ell_{\mathrm{cCPO}}$ | $\log \frac{P_\theta(y_w \mid x)}{(1 - P_\theta(y_w \mid x)) P_\theta(y_l \mid x)}$ |
| $\ell_{\mathrm{qfUNL}}$ | $\log \frac{(1 - P_\theta(y_l \mid x))}{(1 - P_\theta(y_w \mid x))}$ |
| $\ell_{\mathrm{cfUNL}}$ | $\log \frac{(1 - P_\theta(y_l \mid x))}{(1 - P_\theta(y_w \mid x)) P_\theta(y_l \mid x)}$ |
| $\ell_{\mathrm{unCPO}}$ | $\log \frac{P_\theta(y_l \mid x) P_\theta(y_w \mid x) + (1 - P_\theta(y_l \mid x))}{P_\theta(y_l \mid x)(1 - P_\theta(y_w \mid x))}$ |

As described above, the final loss that we implemented was then obtained by applying the logistic loss loss over these equations and adding a $\beta$ term and cross-entropy terms (see details below). We used the `trl` library for implementation from (von Werra et al., 2020), with assistance from the trainer scripts used in Meng et al. (2024).[4]

**Extending the loss lattice to reference models** As seen in Table 2, single model losses can be mapped to DPO-style losses with reference models by subtracting the log ratio $s_{\mathrm{ref}}(y_w, y_l)$ from their loss equation $\rho_\theta$, which we call the **reference form** of a single model loss. We note the following fact about reference loss forms. Formally, given any core loss equation $\rho_\theta$ equal to $\log \rho_\theta^t / \rho_\theta^b$, the reference form of that loss (i.e., $\rho_\theta - s_{\mathrm{ref}}(y_w, y_l)$ with $s_{\mathrm{ref}}(y_w, y_l) := \log P_{\mathrm{ref}}(y_w \mid x) / P_{\mathrm{ref}}(y_l \mid x)$) is equal to the core loss equation $\rho_\theta^{\mathrm{ref}} := \log \frac{\rho_\theta^t P_{\mathrm{ref}}(y_l \mid x)}{\rho_\theta^b P_{\mathrm{ref}}(y_w \mid x)}$, which follows from the application of the quotient rule for logarithms.

As an example, the reference form of $\ell_{\mathrm{CPO}}$ is equal to $\ell_{\mathrm{DPO}}$, given that the reference form of $s_\theta(y_w, y_l)$ (i.e., CPO's loss equation) is $s_\theta(y_w, y_l) - s_{\mathrm{ref}}(y_w, y_l)$ (DPO). Using the quotient rule for logarithms, we can transform this into the core loss equation $\rho_\theta^{\mathrm{ref}}$ equal to $\log \frac{P_\theta(y_w \mid x) P_{\mathrm{ref}}(y_l \mid x)}{P_\theta(y_l \mid x) P_{\mathrm{ref}}(y_w \mid x)}$, which confirms the observation above. In contrast, the reference form of $\ell_{\mathrm{ORPO}}$ is a novel loss $s_\theta(y_w, y_l) - s_\theta(\overline{y_w}, \overline{y_l}) - s_{\mathrm{ref}}(y_w, y_l)$ corresponding, after the same algebraic manipulation, to the new loss shown in Figure 1 (**DPO variant**) and the core loss equation $\log \frac{P_\theta(y_w \mid x)(1 - P_\theta(y_l \mid x)) P_{\mathrm{ref}}(y_l \mid x)}{P_\theta(y_l \mid x)(1 - P_\theta(y_w \mid x)) P_{\mathrm{ref}}(y_w \mid x)}$.

Given how our decompilation procedure works via Algorithm 1, we note the following fact about how such extra reference terms affect the semantics of the original loss:

**Proposition 5** (semantics of reference forms). *Given a loss characterized by the core loss equation $\rho_\theta$ equal to $\log \rho_\theta^t / \rho_\theta^b$, the core semantic formula $\mathsf{P}$ for that loss's reference form is logically equivalent to the formula* $(\mathrm{SEM}(\rho_\theta^b) \wedge \textbf{\textit{Ref}}(\mathsf{x}, \mathsf{y}_w)) \to (\mathrm{SEM}(\rho_\theta^t) \wedge \textbf{\textit{Ref}}(\mathsf{x}, \mathsf{y}_l))$.

Figure 8 (gray boxes) show the semantics of the reference forms for a more exhaustive set of single model losses from Figure 5 using the recipe above. This reveals a wide range of novel variants of DPO that we leave for future experiments and study. Correspondingly, Figure 7 shows the Boolean semantics of DPO/SimPO and some novel variants based on the reference form of ORPO ($\ell_{\mathrm{ORPO-ref}}$), qfUNL ($\ell_{\mathrm{qfUNL-ref}}$) and l5 ($\ell_{\mathrm{l5-ref}}$).

**Computing preference structures** Figure 9 shows how to symbolically compute preference structure representations in Python using the computer algebra library Sympy

---

[4]see https://github.com/huggingface/trl and https://github.com/princeton-nlp/SimPO.

| $\mathbf{M}(x, y_w)$ | $\mathbf{M}(x, y_l)$ | $\ell_{\texttt{ORPO}}$ | $\ell_{\texttt{cUnl}}$ | $\ell_{\texttt{l3}}$ | $\ell_{\texttt{CEUnl}}$ | $\ell_{\texttt{cCPO}}$ | $\ell_{\texttt{CPO}}$ | $\ell_{\texttt{CE}}$ | $\ell_{\texttt{sCE}}$ |
|---|---|---|---|---|---|---|---|---|---|
| T | T | | | ✗ | | ✗ | ✓✗ | ✓ | ✓✗ |
| T | F | ✓ | ✓ | ✓ | ✓ | ✓ | ✓ | ✓ | ✓ |
| F | T | ✗ | ✗ | ✗ | ✗ | ✗ | ✗ | ✗ | ✗ |
| F | F | | | ✗ | ✗ | | | ✗ | ✗ |

| $\mathbf{M}(x, y_w)$ | $\mathbf{M}(x, y_l)$ | $\ell_{\texttt{cfUnl}}$ | $\ell_{\texttt{fUnl}}$ | $\ell_{\texttt{qfUnl}}$ | $\ell_{\texttt{l20}}$ | $\ell_{\texttt{unCPO}}$ | $\ell_{\texttt{l14}}$ | $\ell_{\texttt{bCPO}}$ | $\ell_{\texttt{l5}}$ |
|---|---|---|---|---|---|---|---|---|---|
| T | T | | ✗ | | ✗ | ✓ | ✓✗ | ✓ | ✓✗ |
| T | F | ✓ | ✓ | ✓ | ✓ | ✓ | ✓ | ✓ | ✓ |
| F | T | ✗ | ✗ | ✗ | ✗ | ✗ | ✗ | ✗ | ✗ |
| F | F | ✓ | ✓ | ✓✗ | ✓✗ | ✓ | ✓ | ✓✗ | ✓✗ |

*Figure 6.* A Boolean representation (in the style of Figure 4) of the single model loss functions shown in Figure 5. See again Figure 4 for how to interpret the corresponding losses.

| $(\mathbf{M})\,\mathbf{Ref}(x, y_w)$ | $\mathbf{M}(x, y_l)$ | $(\mathbf{M})\,\mathbf{Ref}(x, y_l)$ | $\mathbf{M}(x, y_w)$ | $\ell_{\texttt{DPO/SimPO}}$ | $\ell_{\texttt{orpo-ref}}$ | $\ell_{\texttt{qfUNL-ref}}$ | $\ell_{\texttt{l5-ref}}$ |
|---|---|---|---|---|---|---|---|
| F | F | F | F | | | | ✓ |
| F | F | F | T | | | | ✓ |
| F | F | T | F | | | ✓ | ✓ |
| F | F | T | T | ✓ | ✓ | ✓ | ✓ |
| F | T | F | F | | | | |
| F | T | F | T | | | | ✓ |
| F | T | T | F | | | | |
| F | T | T | T | ✓ | | | ✓ |
| T | F | F | F | | | ✗ | ✓✗ |
| T | F | F | T | | | | ✓ |
| T | F | T | F | | | ✓✗ | ✓✗ |
| T | F | T | T | ✓ | ✓ | ✓ | ✓ |
| T | T | F | F | ✗ | ✗ | ✗ | ✗ |
| T | T | F | T | ✗ | | | ✓✗ |
| T | T | T | F | ✗ | ✗ | ✗ | ✗ |
| T | T | T | T | ✓✗ | | | ✓✗ |

*Figure 7.* Boolean semantics of DPO and SimPO (column 5) and some novel variants of (columns 6-8) representing the different semantic regions in Figure 8.

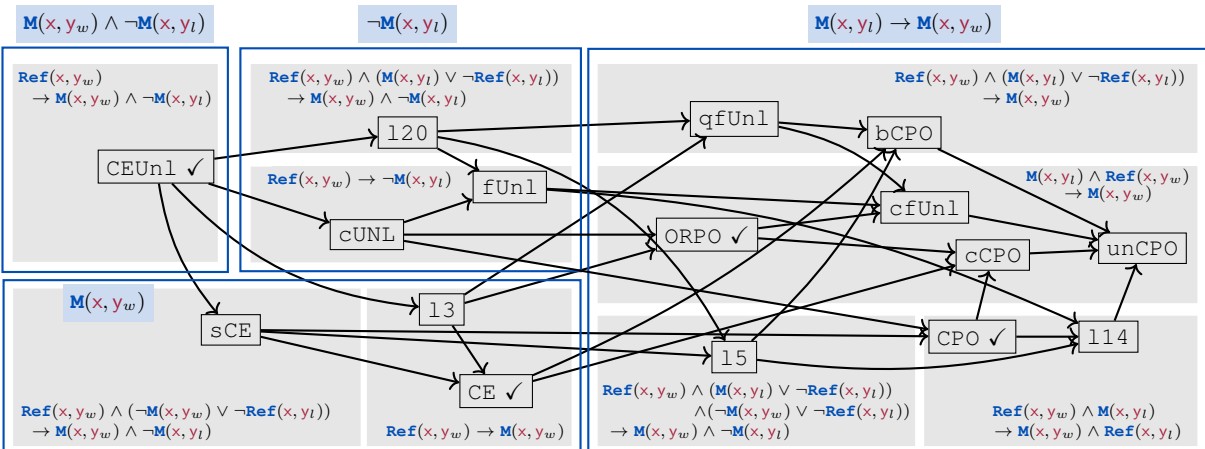

*Figure 8. What are interesting DPO variants to explore?* Extending the loss lattice in Figure 5 to a version of the single model losses with reference models (i.e., their **reference forms**), showing different (largely unexplored) variants of DPO and the different semantics regions (gray boxes, corresponding to the core semantic formula for P each set of losses). See Appendix F for details.

```
1  from sympy import *
2  # winner (W), loser (L),
3  #(ref) winner (R_w), loser (R_l)
4  W,L,R_w,R_l = symbols('W,L,R_w,R_l')
5  ## equation translation for ORPO
6  P_t = And(W,Not(L))
7  P_b = And(L,Not(W))
8  ## pref. structure P̄ = (P,P_C,P_A)
9  P = Implies(P_b,P_t).simplify()
10 assert P.equals(Implies(L,W))
11 P_C = Or(P_t,P_b).simplify()
12 P_A = And(P_t,P_b).simplify()
13 ## The reference form formula
14 P_ref = Implies(
15     And(P_b,R_w), And(P_t,R_l)
16 ).simplify()
17 assert P_ref.equals(
18     Implies(And(R_w,L),W)
19 )
```

*Figure 9.* An example showing how to compute the simplified symbolic formulas in preference structures for `ORPO` (see Figure 2) in `Sympy` (Meurer et al., 2017).

(Meurer et al., 2017). Specifically, lines 8-12 show how to compute a preference structure in the no-reference case, and lines 14-20 show how to compute a reference form of `ORPO` by adding a reference ratio.

## G. Experiments and Case studies

Our formal analysis reveals that the space of DPA losses is large, yet structured in systematic ways that we can now describe through symbolic encodings. Through case studies involving the new losses in Figure 5, we discuss some empirical results that give tips for how to better navigate this space and look for improved DPA losses using our framework. Specifically, we focus on losses around the known loss $\ell_{\text{CPO}}$, which we treat as a natural baseline to compare against. All experiments are performed using a 0.5 billion parameter LLM, `Qwen-0.5B` (Bai et al., 2023), tuned using `trl` (von Werra et al., 2020) on the `ultrafeedback` dataset; following standard practice, losses were implemented with a weighted cross-entropy regularizer term.

While these experiments are small scale and limited in scope, they are merely meant to suggest possible uses our framework and open questions. We also share some general observations and conjectures that we hope motivates future research in this area.

Below we provide details of the experiment setting then discuss some results and observations.

**Dataset and Model**  Following much of the DPA work we cite, we train models on the `ultrafeedback` dataset (Cui et al., 2024), which contains around 60k binarized preference pairs aggregated from several individual preference datasets (the different categories are listed in Table 5). For tuning (detailed below) we used a custom held-out development set containing around 1.3k examples taken from the train set and reserve the test set (containing 2k examples) for final evaluation.

Standardly, we ran experiments starting from a instruction tuned model (SFT), using a `Qwen-0.5B` (containing .5 billion parameters) base model (Bai et al., 2023) that was initially tuned on 6k pairs from the `deita` dataset of (Liu et al., 2024). To avoid repeating the process of instruction tuning, we started from the trained `Qwen` model released in the TRL library[5]. Our full code is available at `https://github.com/allenai/declarative_preference_alignment`.

**Hyper-parameters and model selection**  The following are the standard set of tunable hyper-parameters involved in our experiments: the $\beta$ term for DPA losses (see again Table 1), the learning rate, number of epochs, batch size and length normalization. Following other studies, we also regularized our losses with cross-entropy terms (CE) that include a tunable weight parameter $\lambda$ that controls their contribution to the gradient. Specifically, we kept set $\beta$ to 1, and experimented with learning rates in the range $\{$`1e-6`, `3e-6`, `8e-6`$\}$, number of epochs in the range of $\{3, 5, 8\}$ and batches sizes in the range $\{$ `32`, `128` $\}$ (for efficiency reasons, most tuning with done with a batch size of 32), which follow many of the suggested ranges in Meng et al. (2024). Importantly, length normalization was used throughout to make all losses comparable and given that it has been shown to improve training performance (Meng et al., 2024). We used $\lambda$s in the range of $\{0.0, 0.01, 0.1, 0.3, 1.0\}$ (we found lower values, around $0.01$ and $0.1$, to be most effective).

For each loss function we searched the best hyper-parameters by performing a comprehensive search over the ranges detailed above. Final model selection was then performed by performing inference with each trained model on our held-out development set and scoring the resulting generating outputs using an off-the-shelf reward model, in particular, a 1.8B parameter reward model from (Cai et al., 2024)[6]. We then selected the models with the highest aver-

---
[5] `https://huggingface.co/trl-lib/qwen1.5-0.5b-sft`
[6] `internlm/internlm2-1_8b-reward`

age reward score over the development set for comparison.

For the log probability experiments shown in Figure 10, we kept the learning rate, epoch and cross-entropy term constant (with learning rate equal to `1e-6`, 3 epochs, and a low cross-entropy term $0.01$) to directly compare the different approaches and try to bring out their more extreme behavior.

**Evaluation protocol and win-rate comparison**  We compare models tuned using our different losses using a procedure similar to how model selection is performance, which also follows the setup in Hong et al. (2024). Specifically, we do a instance-level comparison of the reward score given for each generated output, compare that score with the score of our baseline $\ell_{\text{cpo}}$ and compute an overall win-rate, i.e., % of instances where the reward score is higher than or equal to the reward score for $\ell_{\text{cpo}}$ (we consider cases where items are equal given that some tasks involve generating single token output, such as the identifier of a multiple choice question or **yes** or **no**). We report the average win-rate averaged over 3 runs of each models with different generation seeds using `vllm` (Kwon et al., 2023).

### G.1. Results and discussion

**How does constrainedness relate to loss behavior? Unintentional alignment shortcuts**  Moving left to the right in Figure 5 yields semantically less constrained losses. For example, we see through the Boolean semantics in Figure 10 that some unconstrained losses can be satisfied by making the winner and loser both false ($\ell_{\text{unCPO}}, \ell_{\text{cfUNL}}$) or by making the the winner and loser both true ($\ell_{\text{unCPO}}, \ell_{\text{cfUNL}}$). One natural question is: *How does constrainedness contribute to a loss functions empirical success?*

We observe, consistent with other recent work on neurosymbolic modeling (Marconato et al., 2024; van Krieken et al., 2024b), that such unconstrainedness can yield extreme behavior as illustrated in Figure 10. For example, $\ell_{\text{unCPO}}$ and $\ell_{\text{cfUNL}}$ attempt to make both the winners and losers false by driving their probability in the direction of zero (as shown in both training (b) and evaluation (c)), whereas $\ell_{\text{cfUNL}}$ keeps both probabilities high to make both true. When viewing learning as a constraint satisfaction problem, such behavior makes sense and could help to better understand various spurious training behavior observed elsewhere in the DPA literature, e.g., related to likelihood displacement and *unintentional unalignment* studied in Razin et al. (2025) or issues with preference ranking (Chen et al., 2024a).

These results suggest that understanding the way in which a loss is constrained and whether it gives rise to spurious or **unintentional alignment shortcuts** (e.g., making both predictions false) is an important factor when designing new loss functions. We note that existing losses in Figure 5 are in the middle of the two extreme points and seem less

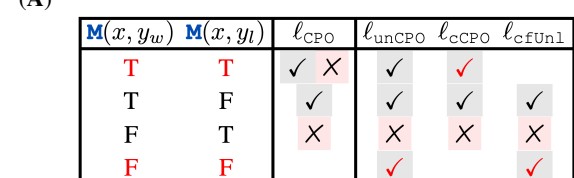

(A)

| $\mathbf{M}(x, y_w)$ | $\mathbf{M}(x, y_l)$ | $\ell_{\text{CPO}}$ | $\ell_{\text{unCPO}}$ | $\ell_{\text{cCPO}}$ | $\ell_{\text{cfUnl}}$ |
|---|---|---|---|---|---|
| T | T | ✓ ✗ | ✓ | ✓ | |
| T | F | ✓ | ✓ | ✓ | ✓ |
| F | T | ✗ | ✗ | ✗ | ✗ |
| F | F | | ✓ | | ✓ |

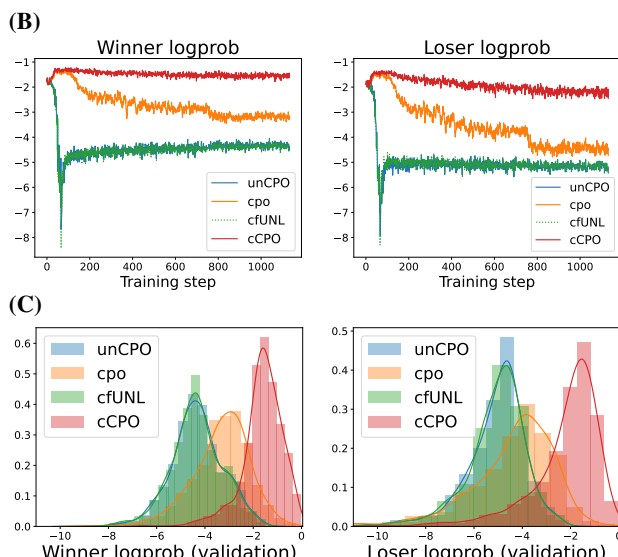

(B)

(C)

*Figure 10.* An illustration (A) of how to semantically satisfy losses ( ✓ ) and the corresponding log probability behavior during training (B) and evaluation (C).

susceptible to such extreme behavior, which could explain their success.

**Can we find empirically improved losses using our method? Formalize and refine**  Our ultimate aim to use our framework to help discover new and successful preference algorithms. Given the spurious behavior of losses $\ell_{\text{unCPO}}$ and $\ell_{\text{cfUNL}}$, we would expect them to be less empirically successful. To test this and compare against $\ell_{\text{CPO}}$, we performed a model-as-judge-style experiment based on (Hong et al., 2024) that uses an off-the-shelf reward model (Cai et al., 2024) to score the outputs generated by our new models using the prompts from the `ultrafeedback` test set. We then compare these rewards scores against those of $\ell_{\text{CPO}}$ to compute a win-rate, which gives an indication of improved or comparable generation quality over $\ell_{\text{CPO}}$. Indeed, we see in Table 5 that in aggregate, $\ell_{\text{unCPO}}$ and $\ell_{\text{cfUNL}}$ have the lowest win-rate against $\ell_{\text{CPO}}$. Interestingly, we see that $\ell_{\text{cCPO}}$ has a win-rate that suggests comparable generation quality to $\ell_{\text{CPO}}$, which shows the potential of using our framework to derive new and empirically successful losses.

These experiments are an exercise in an approach we call **formalize and refine**, i.e., starting from empirically successful losses such as $\ell_{\text{CPO}}$, one can formalize such losses then modify the semantics to be more or less constrained

based on empirical findings. We think more large scale exploration of the full loss space, especially for DPO, is a promising direction of future research.

**Is there a single semantics for all preference learning? The different semantics conjecture** We note that win-rate across different categories in ultrafeedback (i.e., the right most columns in Table 5) varies quite considerably across models and loss types. This suggests that different types of preference data rely on a different semantics of preference, which requires a tuning approach that's tailored to those differences. We conjecture that such a phenomenon is likely to be wide spread across different tasks and datasets, and we see more empirical work on understanding the kinds of semantics needed in different scenarios as a promising direction of future work. Such work will benefit for recent attempts as incorporating more fine-grained annotation into preference such, such as in Miranda et al. (2024).

## H. DPOP equation

The original DPOP loss (Pal et al., 2024) in Table 2 subtracts an additional term $\lambda \cdot \max(0, \log \frac{P_{\text{ref}}(y_w|x)}{P_\theta(y_w|x)})$ from DPO loss, with the aim of ensuring that the log-likelihood of the winner for model $\theta$ is high relative to the reference model (in their study $\lambda$ is set to a whole number ranging from 5 to 50). At first glance, this formulation does not fit the core loss equation format introduced in Table 2. Our approach is the following: first, we make this into a single $\log$ ratio by making $\lambda$ an instance-level parameter set to 0 whenever $\frac{P_{\text{ref}}(y_w|x)}{P_\theta(y_w|x)}$ is less than zero and to be the original $\lambda$ otherwise, resulting in $\log \frac{P_{\text{ref}}(y_w|x)^\lambda}{P_\theta(y_w|x)^\lambda}$. Expanding this out to the full DPOP loss function, this results in the core loss equation:

$$\log \frac{P_{\text{ref}}(y_l \mid x) P_\theta(y_w \mid x)^{\lambda+1}}{P_{\text{ref}}(y_w \mid x)^{\lambda+1} P_\theta(y_l \mid x)}$$

which is not multilinear when $\lambda > 1$. In order to ensure multilinearity, while also maintaining compositionality from Assumption 1, we treat $P_{\text{ref}}(y_w \mid x)^\lambda$ and $P_\theta(y_w \mid x)^\lambda$ as separate probabilistic variables, $P_{\text{ref2}}(y_w \mid x)$ and $P_{\theta2}(y_w \mid x)$, respectively. This results in the core loss equation shown in Tables 2 and 6.

Below we show the core semantic formula for DPOP, which, as noted before, makes a small adjustment to the DPO semantics as shown in Table 4:

```
P := Implies(
    And(Ref(x,y_w), Ref_2(x,y_w), M(x,y_l )),
    And(Ref(x,y_l), M(x,y_w ), M_1(x,y_w ))
)
```

