# OpenReview forum: "Understanding the Logic of Direct Preference Alignment through Logic"
_ICML.cc/2025/Conference — ICML 2025 poster_

### Official Review · Reviewer_f9d6 · 2025-03-13

**Overall Recommendation:** 3

**Summary:**

This paper proposes a symbolic method to interpret direct preference alignment (DPA) loss functions. Given a DPA loss, the proposed method translates it into a preference structure consisting of three formulae, which can be further used to construct a corresponding semantic loss. The paper further shows how existing DPA variants can be converted to semantic loss forms and explores the relation between these different losses. Finally, the paper proposes a simple study about how the formulation can be used to search for new DPA loss functions.

## update after rebuttal

I have read authors' rebuttal and other reviews. I would like to raise my score to weak accept.
The paper can benefit from the promised revisions.

**Claims And Evidence:**

**Claim: **

**Claim: we show how this formal view of preference learning sheds new light on both the size and structure of the DPA loss landscape, making it possible to rigorously characterize the relationships between recent loss proposals.**

This claim is supported by the theoretical analysis of the paper on existing DPA loss and their relations.

**Claim: (the finding also makes it possible) to systematically explore the landscape and derive new loss functions from first principles.**

This claim is not fully supported. Although the paper proposes a limited study in table 5, the found loss term does not exhibit significant advantage over existing ones.

Furthermore, it is unclear to me how the proposed formulation can help explain the performance of different DPA variants, although from figure 4 there are indeed relations and differences in the semantics of these variants.

Furthermore, analysis on DPOP in Section F suggests that ad-hoc treatment is needed for DPA loss with "non-standard" forms. This undermines the generalisability of the proposed method as an explanation and exploration tool.

**Essential References Not Discussed:**

None.

**Experimental Designs Or Analyses:**

There are not many experiments.

Table 5 presents an empirical study on how a new loss, i.e., $L_{cCPO}$, performs compared with existing loss terms. The performance of the found loss is mediocre and the experiment overall does not convince me that this method is able to help with loss design.

**Methods And Evaluation Criteria:**

**Formalising DPA loss terms as semsntic loss**

The method is interesting. However, it seems that the method is limited to the "main" component of the loss, disregarding the regularisation terms. Regularisation terms usually play important roles in those preference optimisation losses. So while I believe this method is able to interpret part of the semantics behind the loss, it may also ignore important aspects.

**Algorithm 1: translation of loss to logic**

The algorithm uses rules in table 6, which covers P1*P2, (1-P1) and P1+P2. I wondered how general this is. For example, how can $\sqrt{P1 \times P2}$ be translated?

**Other Comments Or Suggestions:**

Please refer to my points above.

**Other Strengths And Weaknesses:**

Strengthes:

- This paper proposes a formulation of DPA loss in terms of symbolic representation-based semantic loss. From many existing loss terms, the proposed method draws interesting insights about their semantics.

Weaknesses. Please check the review above, I repeat some important points below.

- The paper lacks empirical or theoretical study on the applicable extend of the proposed method. The translation is based on a set of fixed rules, making it unclear how general these rules are.

- Special treatments are required for some DPA loss variants (e.g., DPOP).

- The paper does not provide many insights on different performances of loss variants, nor does it provide insights on how to design new DPA loss. I wondered how this method can practically benefit preference optimisation studies.

**Questions For Authors:**

Please refer to my points above.

**Relation To Broader Scientific Literature:**

This paper is a symbolic interpretation of preference optimisation losses for value alignment. The topic is important.

**Theoretical Claims:**

I did not check all the theoretical results, but the main results in proposition 2, lemma 1, theorem 2, and Table 4, seem correct.

---

> ### Author Rebuttal · Authors · 2025-04-01
>
> Thank you for your feedback.
>
> > The translation is based on a set of fixed rules, making it unclear how general these rules are.
>
> See comment below about WMC.
>
> > The algorithm uses rules in table 6, which covers P1*P2, (1-P1) and P1+P2. I wondered how general this is. For example, how can  sqrt(P1×P2) be translated?
>
> Given that our analysis is based on WMC, the losses being decompiled must be expressible as the polynomial class defined by WMC; by construction, we restrict this to the *disjoint multinear polynomials* defined on line 348. Except for DPOP (see discussion below), this naturally captures all known preference losses, including those in Table 2, which we emphasize is a fairly comprehensive set of DPA losses (we updated our appendix to include additional DPO variants that fit this analysis), as well as the common loss functions such as cross-entropy and unlikelihood.
>
> The square root clearly does not fit this equation class, so it is not covered by our analysis. Is there a particular loss involving a square root that you have or mind, or some other salient loss that seems out of scope?
>
> > The method is limited to the "main" component of the loss, disregarding the regularisation terms. Regularisation terms usually play important roles in those preference optimisation losses. So while I believe this method is able to interpret part of the semantics behind the loss, it may also ignore important aspects.
>
> This is an important point, which we will say more about in the updated draft. We note, however, that the decision to avoid regularization terms in our analysis follows other formal studies such as by Tang et al. (2024). We also found cross-entropy terms, even when removed, to have limited impact on empirical performance, which is a finding consistent with [1].
>
> [1] Hanyang Zhao et al. RainboPO: A Unified Framework for Combining Improvements in Preference Optimization
>
> > Special treatments are required for some DPA loss variants (e.g., DPOP).
>
> This special treatment is due to the compositional constraint from Assumption 1 (inspired from programming semantics), which stipulates that every model prediction in a loss should be directly accounted for in the semantics. *For clarity*: the DPOP loss has terms such as $p_{\theta}(y_{w} \mid x)^{k}$ and $p_{ref}(y_{w} \mid x)^{k}$ with exponents $k$ that make the equation not multilinear when $k > 1$ (without loss of generality we only considered cases where $k=1,2$).  Our decision to make them multilinear when $k >1$, by creating new variables $p_{\theta2}(y_{w} \mid x)$ and $p_{ref2}(y_{w} \mid x)$ is based on a common polynomial transformation for making polynomials multilinear, which in the end maintains compositionality. Importantly, as we also detail in Appendix F, the value of $k$ also varies from the instance to instance, so both $p_{\theta2}(y_{w} \mid x)$ and $p_{ref2}(y_{w} \mid x)$ do not have a fix value (i.e., they either can take the value of the other variables or be equal to 1 when $k=1$), which we think justifies treating them as semantically distinct values.
>
> We will discuss this more in the draft, including making this more clear when we first reference DPOP around lines 140 (second column). We acknowledge that such exponents could be treated differently, e.g., as parameters that are part of the probability computation for each variable similar to how length normalization is computed, which would sidestep completely these issues involving multilinearity.
>
> > Furthermore, it is unclear to me how the proposed formulation can help explain the performance of different DPA variants, although from figure 4 there are indeed relations and differences in the semantics of these variants.
>
>
> As we discussed in our response to JCXg , Proposition 3 establishes that semantic entailment induces certain monotonicity properties w.r.t to the behavior of the compiled losses, and provides us with a notion of the relative constrainedness of losses that is grounded in loss behavior. As discussed in Sec.E.1, the relative constrainedness of a loss is a property that we think has an important impact on its empirical performance, which is explained by our framework and observable when looking at training dynamics as shown in Figure 8.
>
> We moved this analysis into the appendix due to space issues. If accepted, we intend to include some of this empirical analysis in the main paper (including some additional experiments that scale the reported ones).

---

### Official Review · Reviewer_zia1 · 2025-03-13

**Overall Recommendation:** 3

**Summary:**

This paper introduces a novel approach to describing direct preference alignment (DPA) algorithms in terms of propositional logic. By generalizing the notion of semantic loss, the authors attempt to provide a formal framework to characterize differences between DPA variants.
Subsequently, the authors leverage the introduced framework to discover improved loss formulations.
Experiments on a small (0.5B params) LLM show that the proposed systematic comparison can produce better loss functions for DPA.

**Claims And Evidence:**

The central claim of the paper is that the proposed framework allows for novel insights into relationships between existing DPA formulations. Additionally, they propose to leverage that framework to create new loss formulations in a structured manner.

Given that this is more of a theoretical paper the authors provide limited experiments.

However, they verify their first claim by deriving a loss landscape (or lattice) that provides valuable insights into the interplay between different losses.
Given that landscape they also derive new loss formulation that outperform existing formulations.

**Essential References Not Discussed:**

n/a

**Experimental Designs Or Analyses:**

As discussed above the theoretical analysis is strong and well rounded.
Only one actual experiment is performed, but the setup and choice of evaluation datasets is sound as well.

**Methods And Evaluation Criteria:**

- the theoretical results are strong and reasonably evaluated/demonstrated
- The subsequent empirical experiments are decent but lack some depth since only one (and very small) LLM was considered

**Other Comments Or Suggestions:**

Related to the point above, Appendix E contains many important results to validate the proposed frameworks’ capability. I would encourage authors to include key insights in the main text as well.

**Other Strengths And Weaknesses:**

# Weaknesses
While the paper provides interesting insights the main impact on the field remains somewhat opaque.
For example,
- How well can other researchers leverage the proposed framework to identify novel loss formulations beyond the 4 shown in the paper?
- Does this approach generalize to other models and larger parameter counts

**Questions For Authors:**

- Is there any signficant computational overhead with the new loss formulations? (especially cCPO)
- Can the introduced semantics over loss functions contribute to efficient search of loss functions (e.g. pruning redundant ones)?

**Relation To Broader Scientific Literature:**

The idea of using logic to describe different DPO algorithms is novel.
Although the scheme itself may be largely covered by the original semantic loss paper (i.e. composing logic representations based on NNs outputs), the dedicated arguments and discussions for DPO algorithms would be valuable to the community.

**Theoretical Claims:**

The theoratical claims and proofs appear to be sound but my confidence in that part of the review is low.

---

> ### Author Rebuttal · Authors · 2025-04-01
>
> Thank you for your encouraging feedback.
>
> > How well can other researchers leverage the proposed framework to identify novel loss formulations beyond the 4 shown in the paper?
>
> While we reported experiments on 4 losses, we derived many more novel losses, including the 16 single model losses shown in Figure 7 each (excluding 1) with a novel DPO variant. We think that this set exhaustively captures the full set of single model losses that researchers would want to experiment with (it exhaustively shows all definable losses between CEUnl and unCPO), and offers many novel DPO variants that have not yet received empirical verification. We plan to release all the associated code and loss implementations to facilitate further work in this area.
>
> Given that our framework now allows one to define losses using a much more expressive logical language, where any valid propositional formula constitutes a valid loss function, we also believe that this makes it possible to more easily devise entirely new classes of loss functions that would otherwise be difficult to derive working solely from the mathematics of DPO (e.g., losses that involve more complex forms of feedback, non-differentiable components).
>
> We will include a concrete example of this in the updated draft.
>
> >Does this approach generalize to other models and larger parameter counts
>
> These approaches can be generalized to any language model of any size or number of parameters; the size of the underlying language model is usually not a critical factor in the loss computation (see below).
>
> > Is there any significant computational overhead with the new loss formulations? (especially cCPO)
>
> No. The cCPO loss is computed according to the equation given in Appendix D, and does not require any more computation than another of the other losses in that table. As above, computing losses is a relatively easy computation once the basic forward calls have been made to the model to obtain model output probabilities, which are done independently of the loss computation.
>
> >Related to the point above, Appendix E contains many important results to validate the proposed frameworks’ capability.
>
> Thank you for this suggestion. If accepted we intend to use the additional page to move into the main paper some of the details of the experiments now in the appendix (these were not included due to space limitations), as well as subsequent experiments we did to further verify these findings (see our response to `JCXg`).

---

### Official Review · Reviewer_JCXg · 2025-03-15

**Overall Recommendation:** 2

**Summary:**

This paper presents a fresh perspective on common loss functions in the rapidly growing direct preference optimization literature. In particular, by translating loss functions into symbolic expressions, the paper offers a principled way to analyze their semantics. This approach makes it easier to understand relationships between different DPO-style algorithms, in particular by revealing a hierarchy based on the amount of constraints imposed by each individual algorithm.

**Claims And Evidence:**

The paper makes two claims:

1- the translation into discrete reasoning allows us to better understand the differences between recent proposals in DPO-style algorithms. This is well-supported.

2- the second claim is that this Logic perspective allows us to derive and develop new DPO-style algorithms. While some preliminary signals are presented, this claim is not fully supported, as the evaluation for the new algorithms feel quite limited, on a single benchmark and with limited comparison with existing algorithms and also limited insights found. I think the key question about this paper remains what authors ask on the last page: Can we find empirically improved losses using this method?

**Essential References Not Discussed:**

NA

**Experimental Designs Or Analyses:**

The finding about the large space of algorithms being possible is really exciting. However, while the paper briefly mentions new loss variants, it lacks a strong empirical demonstration of how these losses improve performance over existing algorithms.

**Methods And Evaluation Criteria:**

NA

**Other Comments Or Suggestions:**

The paper states that the number of definable preference structures is doubly exponential in the number of model predictions. While this highlights a rich space of potential loss functions, it also raises concerns about how efficiently we can explore this space. Having read the paper, it is hard to identify a practical approach for navigating this space.

**Other Strengths And Weaknesses:**

My main concern is regarding the notion of using an absolute value of \epsilon for determining "valid" model predictions. This seems somewhat ambiguous. If a winner in one preference pair is a loser in another, how does the framework handle such inconsistencies?

In particular suppose that we have:

x, y_1, y_2
x, y_2, y_3

in our dataset, which in fact can be fairly common. What would be the value of \epsilon in this case?

**Questions For Authors:**

The paper identifies that the number of definable structures is in fact doubly exponential. How do we then navigate such a huge space so that we actually get a practical benefit from this insight?

You mention that the framework could help derive new DPA losses, but the paper primarily formalizes existing ones, with some limited observations about new algorithms. Have you discovered any novel loss functions that outperform existing approaches, and more importantly, a mechanism to understand which new algorithms could be more promising?

Weighted Model Counting (WMC) is a core component of the framework for translating DPA losses into logical expressions. In light of the computational complexity of WMC how does this approach scale? What is the computational bottleneck in algorithm 1?

**Relation To Broader Scientific Literature:**

Instead of focusing on empirical improvements, this work formalizes DPA loss functions using symbolic logic.
The work introduces preference structures, which categorize loss functions using logical relationships rather than just their optimization properties. This builds on the trend of moving beyond black-box RLHF and into more interpretative framework.

**Theoretical Claims:**

NA

---

> ### Author Rebuttal · Authors · 2025-04-01
>
> Thank you, we are excited that you find our work “really exciting”, below we address your comments and concerns.
>
> >However, while the paper briefly mentions new loss variants, it lacks a strong empirical demonstration
>
> Please see our response below. Much of our experimental results were pushed to the appendix due to space limitations.  If accepted we intend to use the extra page to include more empirical results in the main paper, including the extra experiments we mention below.
>
> > computational complexity of WMC
>
> Indeed, WMC is a hard algorithmic problem, but we note that in our study this complexity is side-stepped since we are working with problems with a small number of variables (2-4). In such cases, one can simply write the WMC formulas in full and simplify them using algebra offline, which is how we obtained the new losses in App.D (we updated the draft to include details of how to compute this using SymPy).
>
> Nonetheless, the probabilistic reasoning community has devised various “knowledge compilation” techniques that allow one to scale WMC to much larger problems by compiling these logical representations into tractable circuits, which make many problems feasible in practice (and sometimes in theory). We are confident that if we were to greatly expand the complexity of our programs, such techniques, or other advanced SAT techniques, could be leveraged for doing the involved experiments.
>
> >Have you discovered any novel loss functions that outperform existing approaches, and more importantly, a mechanism
>
> Yes, we found in the experiments reported in Table 5 that the novel loss, $\ell_{cCPO}$ from Figure 4 outperforms the known loss $\ell_{CPO}$ in a win-rate study adapted from Meng et al 2024. This does show evidence that our theory is able to yield competitive new losses (we have subsequently run these same experiments at larger scales using a >3x larger Smollm-1.7B model and are seeing similar trends; we intend to report these results in the updated draft pending the full results).
>
> In terms of mechanisms, we think our experiments do give insight into this, as noted below.
>
> – We find that the logical constrainedness of a loss function is an important contributing factor to its empirical success. For example, the highly unconstrained losses in Fig.4 tend to have spurious behavior due to the nature of their semantics and how the underlying constraints can be satisfied (see discussion in E.1), which we can see clearly in the behavior of their win/lose log probabilities and training dynamics shown in Figure 8 (such empirical behavior is something we consistently see across the many experiments we’ve run at different scales).
>
> Since DPO was introduced, there has been much puzzlement about the empirical behavior of log probabilities during training, which we think our constraint satisfaction view of the problem can help to elucidate.
>
> – We also see in Table 5 that losses perform differently on different subsets of data, which is a trend that we’ve seen persist through our later experiments. We take this as evidence that different tasks and datasets involve different semantics, requiring one to carefully tailor their losses to those semantics.
>
> > How do we then navigate such a huge space
>
> We tried to address this in “How is the loss space structured” (the right column of line 287). While the space is indeed large, we can see through results such as Proposition 3 that loss behavior is linked in interesting ways to the logical semantics of the losses, which can be exploited for exploration. The strategy we pursued in our case study was the following: start with an empirically successful loss (e.g., CPO), formalize its semantics, then modify its systems to find new losses that are either more constrained (or that entail CPO) or less constrained (that are entailed by CPO) and experiment accordingly.
>
> We think strategies like this are useful tools for navigating this space.
>
> > My main concern is regarding the notion of using an absolute value
>
> The \epsilon notion is only meant to be a tool or heuristic to conceptually think of the distribution as digitized or divided into “valid” and “invalid” outputs; it is not a value that we model explicitly or that plays any role in our formal analysis, and hence we do not make any assumptions about it (e.g., it needn’t be a fixed value).
>
> Regarding your example: if we interpret the top symbolic formula in Figure 2 in terms of this digitized distribution, this just says that “whenever we find the loser to be a valid generation (i.e., to be above this \epsilon) we should always find the winner to also be above this line (\epsilon) too”. Note that this semantics, which underlies many DPA approaches, does not rule out the possibility that the loser is a “valid” generation (or also above \epsilon). So for your example of `(x, y_1, y_2) (x, y_2, y_3)` we could set this hypothetical \epsilon to be the probability `y_3` while still satisfying this constraint.

---

### Official Review · Reviewer_pLhs · 2025-03-18

**Overall Recommendation:** 4

**Summary:**

This paper proposes a novel framework to unify various preference optimization losses as a logical program of response orders. A pairwise preference implies the logic that a rejected response in the policy shall imply that the chosen response is in the policy; A supervised loss implies that the response is favored by the policy.

The proposed framework characterizes each preference optimization loss function as such as logical program. On the other hand, they provide an mechanism to remap a logical program to a corresponding loss function. By exploring in the space of logical implications, they essentially explore in the space of optimization losses.

In addition to drawing the relationship between losses and logical constraints, their framework provides an interesting roadmap to understand the relationship between different losses.

Finally, they provide a proof-of-concept experiment showing that their approach has the potential to discover better preference optimization losses.

**Claims And Evidence:**

This paper is mostly a theoretical paper that provides a mathematical framework between preference optimization loss and logical implications. I have followed most of its derivations in the paper, which look good to me.

**Essential References Not Discussed:**

Not necessarily essential references, but there have been many relevant papers which incorporate "scalar reward signals" in the training objective. With the fine-grained reward information, those methods tend to outperform methods relying on binary preference labels.

It would be a strong addition if this framework could include these methods into the roadmap as well. Some relevant papers are follows.

[1] RPO. Nemotron-4 340B Technical Report;
[2] Distill DPO. Robust Preference Optimization through Reward Model Distillation.
[3] InfoNCA. Noise Contrastive Alignment of Language Models with Explicit Rewards.
[4] Brain: Bayesian reward-conditioned amor- ´ tized inference for natural language generation from feedback.

**Experimental Designs Or Analyses:**

This paper involves a minimal proof-of-concept experiment only. If there are more compelling experiments showing some really strong losses, it will help the paper more.

**Methods And Evaluation Criteria:**

The proposed method is a brand new viewpoint of preference optimization losses. The most interesting finding is that most of the preference optimization losses are a combination of "semantic log ratios". Each semantic log ratio corresponds to a certain logic constraint, which are then combined together in the overall loss function. It is a very interesting framework to shed more light on understanding preference optimization algorithms.

Beyond its main mathematical contributions, the paper shows a proof-of-concept experiment demonstrating some new losses found in their framework. While the experiment is only minimal, this framework provides the potential to dynamically search for the best loss function, similar to how neural architecture search improves neural network performances.

**Other Comments Or Suggestions:**

None.

**Other Strengths And Weaknesses:**

This paper is well written. The logic is easy to follow.

**Questions For Authors:**

None

**Relation To Broader Scientific Literature:**

This paper is a valuable addition to the relevant science literature. In the literature, the preference optimization landscape has been very complicated, due to the combination of many design factors like loss function and reference policy. This paper provides a novel interpretation for those algorithms and a roadmap to connect the scattered dots.

**Theoretical Claims:**

I can follow the logical flows in the paper. But I didn't check the proofs carefully.

---

> ### Author Rebuttal · Authors · 2025-04-01
>
> Thank you for your feedback, we are pleased that you find our approach to be “a valuable addition to the relevant science literature”
>
> >many relevant papers which incorporate "scalar reward signals" in the training objective. With the fine-grained reward information, those methods tend to outperform methods relying on binary preference labels.
>
> Thank you for these pointers. This does look like an exciting new area where we can try to apply our techniques. While we haven’t digested yet all the details of these papers, it does seem like the kind of distillation objective in Fisch et al. 2025 (Eq. 7) could fit into our framework by adding these additional reward model estimates into our logical formulas as additional predicates (we will think more about this and, if appropriate, mention this direction in an updated draft).

---

### Official Review · Reviewer_mbGz · 2025-03-19

**Overall Recommendation:** 2

**Summary:**

The paper proposed the decompilation of loss functions such as DPO into symbolic programs. More specifically, the authors present how to derive probabilistic propositional logic programs that can, in turn, be manipulated and compiled into potentially novel and improved losses for preference alignment.

**Claims And Evidence:**

The paper claims to (i) introduce formal insights into and characterization of DPA losses through their decompilation into symbolic programs and (ii) practical insights into effective searches for novel DPA losses that improve over the state-of-the-art. Although compelling, the work does not convincingly support the claims, as it lacks clarity in the formal presentation and only provides minor practical insights on feasibility.

**Essential References Not Discussed:**

Essential references have been discussed.

**Experimental Designs Or Analyses:**

Compared to the paper's claims, I found the experiment design lacking in its scope and discussion. Although the authors state "While these experiments are small scale and limited in scope, they are merely meant to suggest possible uses our framework and open questions." (page 14, line 716), the experiment mostly shows a basic usage example without providing deeper insights into the method or discussion on limitations.

**Methods And Evaluation Criteria:**

The proposed method seems plausible to improve preference alignment techniques through the interpretation and manipulation of losses on a symbolic level. However, further experiments and discussions are required to evaluate the approach fully.

**Other Comments Or Suggestions:**

- On page 3, line 146, "No reference" is in bold text, this might be by mistake.
- Figure 1 illustrates the core idea of the paper well, but the concepts within the symbolic program may be difficult to parse for a first-time reader on page 1. Perhaps additional annotation or a simplified illustration/example would be more digestible at this point in the paper.
- The abstract employs the abbreviation DPO without introducing it beforehand.
- The use of bold text for both sectioning and underlining important concepts/keywords may be suboptimal.

**Other Strengths And Weaknesses:**

Strengths:
- The paper introduces a compelling method for the decompilation of loss functions into probabilistic propositional logic (and vice versa), opening an interesting avenue for explaining, manipulating, or exploring losses.
- The research direction of the paper is important and approached in an interesting and original way.

Weaknesses:
- The paper contains some confusing wording that needs to be improved. For example, at the end of 'Neuro-symbolic modeling,' the authors first state, "In particular, we focus on approaches based on probabilistic logic.". The next sentence disagrees, "In contrast, we focus on the inverse [...]".
- Some figures, e.g., Table 2, are visually challenging to parse, so I recommend reworking them for clarity. Others, like Tables 1, 3, and 4, are less problematic but may similarly be improved.
- Some statements made in the paper need clarification. For instance, on page 3, the authors write, "We use θ2 and ref2 to refer
to copies of our two models, which is a decision that we address later [...]". At this point in the paper, the statement left me somewhat puzzled in its meaning.
- The paper's presentation suffers from how information is distributed throughout the manuscript. For example, on page 5, "Decompilation into semantic loss" is described but references/requires insights from Table 2 (page 3), Section 5.2 (page 7), and Table 6 (page 11) to be understood.
- Along the lines of the previous comments, the paper could benefit from some restructuring. As an example, rather than leading into the experiments, Section 6 "Results and Discussion" begins with two more theorems, only one of which is followed by a proof paragraph.

**Questions For Authors:**

I have no questions for the authors.

**Relation To Broader Scientific Literature:**

This work explores the decompilation of losses employed in preference alignment into probabilistic propositional logic. Hence, it relates to literature aiming to guide the training of deep models such as LLMs to align with some additional human preferences. Furthermore, the chosen representation of the decompiled losses relates to probabilistic logic programming (and thereby methods of statistical relational and neuro-symbolic AI), employing weighted model counting over the automatically generated symbolic representations of the loss.

**Theoretical Claims:**

Although the paper contains theorems (one of which is followed by a proof paragraph, the other is not), the paper overall is lacking in clarity, which hinders thorough checks on the contributions' correctness.

---

> ### Author Rebuttal · Authors · 2025-04-01
>
> Thank you for the feedback; as noted in our response to the above reviewer, we’ve already taken steps to improve the presentation, which we hope will also address your concerns.
>
> We are encouraged that you nonetheless found our approach to be “compelling” and an “interesting avenue to explaining, manipulating, or exploring losses”
>
> Below we address your particular comments and questions.
>
> > the experiment mostly shows a basic usage example without providing deeper insights into the method or discussion on limitations.
>
> Our main contribution is to provide clarity into the ever-growing space of preference losses. The symbolic view of these losses allows us to make the preferences encoded by them explicit, understand relationships between them, and, via counting arguments, enumerate the space of all possible losses with or without a reference model. Moreover, it also provides a conceptual framework for inventing novel kinds of losses.
>
> We also note that much of our experimental results were pushed to the Appendix due space issues (see please our response to JCXg). If accepted, we intend to use the additional page to incorporate such results in the main paper.
>
> > "In particular, we focus on approaches based on probabilistic logic.". The next sentence disagrees, "In contrast, we focus on the inverse [...]".
>
> Sorry for the confusion, we will modify this for clarity.
>
> To clarify: the point about this “inverse problem” is that while most people in the neuro-symbolic field have focused on the problem of “compilation” (i.e., translating symbolic formulas into loss functions by interpreting those formulas using probabilistic logic), we focus on the “inverse” and more unique problem of “decompilation” (i.e., deriving symbolic formulas for *existing* loss functions that we also interpret in terms of probabilistic logic).
>
> In both cases, probabilistic logic is used as the ingredient that makes communicating between these symbolic forms and losses possible, which is why we say that our approach is “based on probabilistic logic” (we felt that this was important to clarify since other popular modes of translation, such as fuzzy logic, could have been used here as an alternative to the probabilistic approach).
>
> >On page 3, line 146, "No reference" is in bold text, this might be by mistake.
>
> We will fix this inconsistency.
>
> > Figure 1 illustrates the core idea of the paper well, but the concepts within the symbolic program may be difficult to parse for a first-time reader on page 1
>
> As noted in the response to **fMQw**, we completely revamped this figure to make it more clear along the lines that you suggest.
>
> >The abstract employs the abbreviation DPO without introducing it beforehand..
>
> Thank you for catching this, we will change DPO to “Direct Preference Optimization”.
>
> > Figure 1 illustrates the core idea of the paper well, but the concepts within the symbolic program may be difficult to parse for a first-time reader on page 1. Perhaps additional annotation or a simplified illustration/example would be more digestible at this point in the paper.
>
> Please see our response above.
>
> > The paper's presentation suffers from how information is distributed throughout the manuscript. For example, on page 5, "Decompilation into semantic loss" is described but references/requires insights from Table 2 (page 3), Section 5.2 (page 7), and Table 6 (page 11) to be understood.
>
> We updated our draft to account for these structural issues.

---

### Official Review · Reviewer_fMQw · 2025-03-24

**Overall Recommendation:** 3

**Summary:**

The work attempts to structure the corpus of existing and discover new optimization losses for direct preference alignment (DPA). To this end, existing DPA methods are unified and cast as a reasoning problem. Namely, each loss corresponds to a set of logic formulas that are optimized via weighted model counting (WMC) under extended preference structures. Logical entailment of the formulas leads to a lattice of losses and possibly new losses where first empirical evaluations are promising.

## update after rebuttal

I want to thank the authors for their insightful and comprehensive response. All major concerns have been resolved. However, the clarity is still not quite where it could be, for which reason I maintain my score of 3 (weak accept).

**Claims And Evidence:**

The main claims are that most common DPA losses can be cast as optimizing probabilistic logic formulas. This is appropriately substantiated by the presented constructions and proofs (see also "Theoretical Claims"). However, it is unclear which of the many DPA methods are covered by the new perspective introduced in the work.

**Essential References Not Discussed:**

I am not aware of any.

**Experimental Designs Or Analyses:**

The experiments are very minimal, yet sufficient for showing the basic feasibility of the approach. A lot of future work could extend this.

**Methods And Evaluation Criteria:**

The presented empirical findings are but a first glimpse of what can be explored, but sufficient to show that the approach is fundamentally feasible.

**Other Comments Or Suggestions:**

Comments on clarity and minor opportunities for improvement:
- To me, the start of Sec. 3 is harder to read than necessary. It might help to introduce the role of $\beta$ a bit earlier and move the information from the caption of Tab. 2 ("All losses ...") to the main text to make it self-contained.
- Fig. 1: DOP2 could also be named as such on the right-hand side.
- Fig. 1 talks about "compilation" and "derivation", while other sections, such as 4.1. (any others), talk about "compilation" and "decompilation" instead. This should be unified.
- Fig. 3: The entire figure is a bit unclear, and it personally confused me more than it helped. It might make sense to remove it, as WMC is well-known and can be learned about from other resources. Otherwise, why does $\checkmark$ both correspond to $P$ and tp $\bar{P_f}$, and similarly for the inverse. The meaning of empty cells in the table is also unlcear to me.
- Some columns in Tab. 5 are missing highlights.
- Using page 16 for a single `)` is avoidable.

**Other Strengths And Weaknesses:**

The presented work uses well-known tools from different sub-disciplines (logic/neuro-symbolic reasoning) to structure the jungle of DPA losses. This very original and promising approach leads to an improved understanding of existing works and possibly new principled discoveries. It is original, yet the writing could be clearer in various places (see, e.g., the next section).

The lack of clarity regarding when the method is applicable, and the description and verification of the method itself are the main weaknesses of the work.

**Questions For Authors:**

- **Q1**: In Sec. 4, l. 172ff, you state, "We assume that all preference loss functions have an internal logic that can be expressed in the form described above." To what degree is this a limitation of the approach? What happens if that is not the case and we apply the translation regardless? Generally, can you more clearly state the conditions needed to apply the translation from loss to your symbolic representation?
- **Q2**: A crucial piece for the understanding of the translation is missing: How is, *intuitively*, the implication of the left-hand side of, say Fig. 1, realized by the loss formula? Some discussion along the lines of "if $\pi_\theta(x, y_w)$ is high, $\sigma$ saturates, and then ...". This would help to understand the compilation/derivation steps and serve as a good sanity check of the method/as a case study.

**Relation To Broader Scientific Literature:**

I cannot judge the completeness, yet I did not find any missing spots.

**Theoretical Claims:**

The paper contains various definitions, theorems, and proofs of them. The definitions are mostly given inline in the text, which is legitimate but slightly hinders clarity.

---

> ### Author Rebuttal · Authors · 2025-04-01
>
> Thank you for your feedback, we are encouraged that you found our approach to be “very original and promising”. We have already taken steps to improve the presentation in the places you mention and we will address your particular points below.
>
> > To me, the start of Sec. 3 is harder to read than necessary. It might help to introduce the role of  β a bit earlier and move the information from the caption of Tab. 2 ("All losses ...") to the main text to make it self-contained.
>
> We will address this in the revised draft.
>
> > Fig. 1: DOP2 could also be named as such on the right-hand side.
>
> We have already modified Figure 1 according to your suggestions and in addition: 1) tried to make the figure easier to read and less blurry and; 2) we expanded the caption to more clearly state the problem and the goals of the paper.
>
> > Fig. 1 talks about "compilation" and "derivation"
>
> Thank you for noticing this. We will fix it to be more consistent.
>
> Also, we restructured part of Section 4 to make clearer the precise relationship between compilation and derivation (i.e., that we treat the former as the inverse of the latter), which we hope better motivates why we immediately start in Sec. 4.1 by talking about the “Compilation to semantic loss”.
>
> > Fig. 3: The entire figure is a bit unclear, and it personally confused me more than it helped. I
>
> As with Figure 1, we did a complete revamp of this Figure, which we hope addresses your confusion and makes the modifications you suggest.
>
> To state the role of Figure 3 more clearly (as we do in our updated draft): Preference structures, via Prop. 2, can be equivalently expressed in terms of *two* Boolean functions, $P_w$ and $P_l$, which correspond to the checkmarks and xmarks in the figure respectively (and are also equivalent to the formula forms in Eq. 4). We think that this view is helpful for: 1) understanding the corresponding model counting problem visually; 2) understanding how our generalized formulation of semantic loss, as captured by the equation in this figure, is general enough to capture things like conditional probabilities (i.e., not satisfying $P_C$, as captured by the white boxes in Fig 3; we will update the arrow from $P_C$ in Fig 3 to point to such white boxes without any checkmark or xmark).
>
> > "We assume that all preference loss functions have an internal logic that can be expressed in the form described above." To what degree is this a limitation of the approach? What happens if that is not the case and we apply the translation regardless?
>
> To clarify this claim, which is quite general, we assume that all preference losses have *some* internal logic that can be expressed in a discrete form. Whether or not they can all be expressed in the logic we propose is a separate question (e.g., it might be that some preference losses outside of our study require a more complex logical system beyond propositional logic. We will clarify this point, but we think it is an important working assumption to note).
>
> For the preference losses under consideration in Table 2 (which cover many of the most popular DPA losses), Thrm. 2, however, does establish that our decompilation procedure and logic correctly captures the logic of these losses in the following sense (which we will make more clear): our decompilation procedure can take any of these losses as input and produce a semantic representation that can be compiled back into exactly and uniquely that loss via our logic.
>
> The main condition that needs to be met for the decompilation procedure to be applicable is that the input loss equation is a “disjoint multilinear polynomial” as defined in line 346 (regarding your other point, we see that a non-inline version of this would be helpful here to make this point more clear). Of course, if the input loss does not follow this polynomial form the translation runs the risk of not being correct, but such losses are out of scope for this study and we could imagine making the translation more complex to expand this to other polynomial classes.
>
> >How is, intuitively, the implication of the left-hand side of, say Fig. 1, realized by the loss formula?
>
> Yes, we agree that a clear intuition here would be helpful. Given the way that our implication construction works (i.e., the construction in the proof of Prop.2 which drives Algorithm 1 and our decompilation procedure), the left side of the implication in semantic representations like those in Figure 1 (or the representations in Table 4) corresponds to the lower/bottom part of the log ratios in the preference losses (Table 2, $\rho_{\theta}^{b}$) and the right side to the upper part of these log ratios ($\rho_{\theta}^{t}$).
>
> > The definitions are mostly given inline in the text
>
> We will fix this, particularly by considering making some of the core definitions that our formal results rely on (non-inlined) formalized definitions.

---

### Decision · Program_Chairs · 2025-05-01

**Decision:**

Accept (poster)

**Comment:**

The paper introduces a formal framework to analyze and understand Direct Preference Alignment (DPA) algorithms, such as Direct Preference Optimization (DPO), which are used to align large language models (LLMs) with human preferences. The authors propose a novel formalism, a kind of logic for preference modeling, that characterizes preference losses for single- and reference-model-based approaches by deriving symbolic expressions that encapsulate their semantics. This enables a systematic comparison of different DPA loss functions as well as the derivation of new loss functions. By formalizing DPA losses in terms of discrete reasoning problems, the paper sheds light on the structure and relationships within the DPA loss landscape. Overall, the paper is very well written and presented. The reviews are a bit mixed with 3x weak reject, 2x weak accept, 1x accept, i.e., a borderline overall. However, two of the reviews were asked as emergency reviews, which I will factor in. One of the weak rejects says "fresh perspective on common loss functions" while it "lacks a strong empirical demonstration" of finding new loss variants. Some of them can be found in the appendix.  The other weak reject also states " The topic is important." So overall, while there are indeed some downsides, overall the reviewers agree that the work is interesting and important and lean towards the accept side. Consequently, and since many of the issues raised have been addressed, I vote for accept.